# Verification of the Implicit World Model in a Generative Model via Adversarial Sequences

**András Balogh[1] and Márk Jelasity[1,2]**
[1]University of Szeged, Hungary  [2]HUN-REN–SZTE Research Group on AI, Hungary
{abalogh,jelasity}@inf.u-szeged.hu

## Abstract

Generative sequence models are typically trained on sample sequences from natural or formal languages. It is a crucial question whether—or to what extent—sample-based training is able to capture the true structure of these languages, often referred to as the "world model". Theoretical results indicate that we can hope for soundness at best, that is, generating valid sequences, but not necessarily all of them. However, it is still important to have practical tools that are able to verify whether a given sequence model is sound. In this study, we focus on chess, as it is a domain that provides enough complexity while having a simple rule-based world model. We propose adversarial sequence generation for verifying the soundness of the sequence model. Our adversaries generate valid sequences so as to force the sequence model to generate an invalid next move prediction. Apart from the falsification of soundness, this method is also suitable for a more fine-grained analysis of the failure modes and the effects of different choices during training. To demonstrate this, we propose a number of methods for adversarial sequence generation and evaluate the approach on a large set of chess models. We train models on random as well as high-quality chess games, using several training recipes. We find that none of the models are sound, but some training techniques and dataset choices are able to improve soundness remarkably. We also investigate the potential application of board state probes in both our training and attack methods. Our findings indicate that the extracted board states have no causal role in next token prediction in most of the models.

## 1 Introduction

Generative sequence models like large language models see increasingly more use in areas where a solid understanding of complex concepts and interactions is critical for their success (Lin et al., 2023; Nijkamp et al., 2023; Li et al., 2022b). Recent findings suggest that such important capabilities might naturally emerge during training, yet our understanding of how this knowledge is represented and used by models is still rather limited (Zheng et al., 2024; Schaeffer et al., 2023).

An interesting aspect of this problem is whether the emergent capabilities of generative models are based on some representation of a system of world-states and transitions, and if so, whether this implicit world model is consistent with reality (Vafa et al., 2024). In order to study implicit world models, recent works proposed the use of synthetic tasks like board games that can be described by formal languages, where the world model is explicitly known (Li et al., 2023; Toshniwal et al., 2022). We can then compare the behavior of generative models to the true world model.

However, it is difficult to test if the implicit world model of the generative model is *sound*, that is, whether it adheres to the true world model. To tackle this problem, we propose a novel methodology based on *adversarial sequence generation*, where an adversary generates valid sequences with the aim of forcing the model to break the formal rules of the true world model.

We examine generative sequence models in the domain of chess, using diverse datasets and various training recipes that facilitate learning the true world model. Our methodology reveals a low level of soundness across the board, along with numerous novel insights into the (lack of) causality of world

state probes, the roles of different training objectives, and the impact of dataset choice, such as size and semantics.

Our contributions are as follows [1] :

- We present a novel adversarial framework for measuring the soundness of implicit world models and evaluate it in the domain of chess.
- We perform a large-scale empirical study. We introduce several training schemes that facilitate learning the true world model over datasets of varying sizes and qualities.
- We analyse the models with our adversarial methodology, and show that none of them are sound, but the choice of training recipe, dataset, and adversary has significant effects.
- We examine the role of linear board state probes during training and evaluation, and find that they have a limited causal connection with the predictions of the models.

### 1.1 RELATED WORK

Instead of explicit generative world models (Ha & Schmidhuber, 2018; Zeng et al., 2023), our paper focuses on implicit world models learned by generative models (Li et al., 2023; Vafa et al., 2024).

One way to verify the soundness of these internal world models is to extract them via mechanistic– (Bereska & Gavves, 2024; Nikankin et al., 2025), or conceptual interpretability methods (Patel & Pavlick, 2022). Of the latter, linear probing (Alain & Bengio, 2017; Hewitt & Manning, 2019) has been used extensively to decode learned concepts from feature representations (Abdou et al., 2021; Li et al., 2021; Hewitt & Liang, 2019; Feng et al., 2025). However, the causal role of board states extracted with probes is not evident, as the main goal of probes is analysing if, and how information is encoded, rather than how it is used (Belinkov & Glass, 2019; Belinkov, 2022). Therefore, probes might incorrectly indicate the soundness of implicit world models (Vafa et al., 2024).

Another approach to verification is to compare the outputs of the model with a formal structure that defines the true world model, such as automata (Liu et al., 2023; Laufer & Kleinberg, 2025), or formal rules (Sun et al., 2024; Wolfram & Schein, 2025). Vafa et al. (2024) propose a framework based on sequence-level distinctions to evaluate whether a language model learned the automaton of the true world model, and show that generative models fail to do so. We extend this line of work with a novel approach to implicit model verification that does not rely on sensitive threshold parameters to define the generated language.

Board games have been extensively used in evaluating the emergent capabilities of language models (Karvonen et al., 2024). Li et al. (2023) successfully train language models on Othello transcripts, and they, along with Nanda et al. (2023) and Hazineh et al. (2023) argue that linear board state probes have causal connections to the model's function, while jylin04 et al. (2024) show that the implicit world model of OthelloGPT is fragmented. Toshniwal et al. (2022) and Karvonen (2024) train language models on chess transcripts and argue through output-based and probing methods that these models have emergent world models that are consistent with the true world model.

## 2 PRELIMINARIES AND NOTATION

Informally speaking, we assume that there is a ground truth world model, and we train a sequence model based only on action sequences generated by this world model. Starting from a (hidden) initial state, an action sequence is recorded by following legal state transitions allowed by the possible actions in the world model. We then ask whether the implicit world model learned from a set of action sequences is consistent with the ground truth world model. Let us elaborate on this setup more formally and present an application as well: the game of chess.

### 2.1 WORLD MODELS AND GENERATIVE MODELS

Let $\Sigma$ be the finite set of all actions in a world model. Let $s = a_1..a_k$ be an action sequence, where $k \geq 0$ and $\forall_i a_i \in \Sigma$. Let the set of all possible action sequences be denoted by $\Sigma^*$.

---

[1]Our code, models, and datasets are available at https://github.com/szegedai/world-model-verification

We assume that the true world model is given through the function $W$, where $W(a_1..a_k) \subseteq \Sigma$ is the set of *valid continuations* of the sequence $a_1..a_k$. We say that a sequence $a_1..a_k$ is *valid* if and only if $a_i \in W(a_1..a_{i-1})$ for all $0 < i \leq k$ (by definition, the empty sequence (i.e. $k = 0$) is valid).

Given a set of valid sequences, we can train a generative model $M : \Sigma^* \rightarrow \Delta(\Sigma)$, which is a model that predicts a probability distribution over $\Sigma$, given an action sequence. Let $M(a|s)$ denote the conditional probability assigned to $a \in \Sigma$ by the model, given $s \in \Sigma^*$. When generating a sequence, we need a decoding policy $m : \Sigma^* \rightarrow \Sigma$. For example, the greedy decoding policy is $m(s) = \arg\max_a M(a|s)$.

Note that it is possible that one action is represented by a sequence of two or more *tokens*, in which case model training and prediction should be understood at the token level.

**Definition 2.1.** A generative model $M$ with decoding policy $m$ is *sound* with respect to the true world model $W$ if and only if for any sequence $s$ that is valid in $W$ and $W(s) \neq \emptyset$, we have $m(s) \in W(s)$.

**Focusing on soundness.** Our problem formulation focuses on the verification of soundness, that is, examining whether the generative model generates only valid sequences. Our method will be able to disprove soundness by searching for counterexamples in the form of valid sequences, for which the sequence model predicts invalid continuations. We note that sound and complete generative models (ones that are identical to the world model) are theoretically impossible to learn from samples, even for regular languages (Gold, 1967) while sound models are at least theoretically possible under reasonable assumptions (Kleinberg & Mullainathan, 2024).

**Scope.** In our formulation, we define $W(s)$ as the valid continuations of $s$ without any restrictions on the complexity of the true world model in question. As a result, our framework generalizes to settings where the true world model is more complex (e.g., a pushdown automaton), as opposed to the framework of (Vafa et al., 2024), which requires the true world model to be a deterministic finite automaton.

## 2.2 CHESS NOTATION

We focus on the game of chess due to its clear and deterministic set of rules that form a ground truth world model of the type introduced above.

Like Toshniwal et al. (2022), we use the Universal Chess Interface (UCI) notation to represent actions (moves). This notation combines the starting and destination squares to represent a move. For example, the notation `e2e4` means the player moved the piece on `e2` to `e4`. Special moves and events (e.g., castling, check, and checkmate) are not explicitly encoded, with the exception of promotion, where the piece type the pawn is promoted to is indicated at the end of the move. For example, the notation `a7a8q` means the pawn on `a7` was moved to `a8` and got promoted to a queen.

## 2.3 BOARD STATE DECODERS

To analyze the soundness of implicit world models, some of our algorithms rely on a board state decoder $B$ that is implemented as an extra head added to a generative model $M$, and trained to predict the current board state $B(M, s)$ from a hidden representation within $M$ after a sequence of moves $s$. Most often, the decoder is a simple linear probe Alain & Bengio (2017) that solves a 13-class classification problem for each of the 64 squares on the board independently, where the classes represent the six piece-types for the two sides, and the empty square.

We will also use the loss function $\mathcal{L}_B(M, s)$ in some of our algorithms that measures the error between the true board state after move sequence $s$ and the predicted board state $B(M, s)$.

## 3 ADVERSARIAL VERIFICATION OF SOUNDNESS

Our goal is to evaluate whether a generative model generates only valid sequences (i.e., its implicit world model is sound) and, if not, we are also interested in the extent of the inconsistency.

**Sequence-level evaluation is essential.** It has been argued by Vafa et al. (2024) that simple metrics like next-token prediction accuracy are misleading because even completely wrong models might have high accuracy. Therefore, there is a need for sequence-level analysis and metrics. Our approach is based on generating *valid but adversarial* sequences such that the generative model predicts an invalid continuation for the sequence.

**Advantages of adversarial verification.** While Vafa et al. (2024) propose a theoretically motivated methodology to verify and quantitatively characterize soundness, their approach requires the definition of the formal language generated by the generative model, which in turn requires an ad hoc probability threshold parameter. At the same time, the adversarial sequences simply seek to provide existential proof that the generative model is incorrect, avoiding the need for defining the generated formal language exactly. However, as we will demonstrate, the method still offers quantitative metrics and a detailed insight into the failure modes of different models through the fine-grained analysis of successful attacks.

### 3.1 THE ABSTRACT ADVERSARY

The key component in our framework is the adversary, whose goal is to force the generative model to generate an invalid next action. It is very important that the adversary itself *will always produce valid sequences*, but in a way so that the next action predicted by the attacked model is invalid.

While an adversary could check all valid sequence prefixes in $W$ up to some length to disprove the soundness of the generative model, this would not be efficient, or even plausible in most cases. Instead, given a sequence prefix $a_1..a_k$ that is valid in $W$, our adversary extends the sequence with $a_{k+1}^*$ based on solving

$$a_{k+1}^* = \underset{a_{k+1} \in W(a_1..a_k)}{\arg\max} f(M, a_1..a_k a_{k+1}), \tag{1}$$

where $f$ is an auxiliary function that attempts to capture, for example, the uncertainty or incorrectness of the sequence model before or after the sequence $a_1..a_k$ is extended with $a_{t+1}$. Note that the maximization is done only over valid actions, so the function $f$ itself does not capture validity, only an order of preference. We will describe multiple design choices for $f$ in Section 3.2.

The attack is successful if, for some index $j > k$, the adversary can force an invalid next action. That is, the adversary finds a sequence $s^* = a_1..a_k a_{k+1}^*..a_j^*$ where $W(s^*) \neq \emptyset$ and $m(s^*) \notin W(s^*)$.

**Two-player games.** Since chess is a two-player game, we adapt our attack framework accordingly by having the adversary play against the sequence model. The attacker always plays with white, so for all $i \geq 1$, the move $a_{2i-1}$ is given by Equation 1, and $a_{2i} = m(a_1...a_{2i-1})$. The attack is successful if, for some $i \geq 1$, $a_{2i}$ is an illegal move.

### 3.2 ADVERSARY IMPLEMENTATIONS

First, we present three attacks, that is, three different implementations of $f$ in Equation 1. We then add two non-adversarial baselines as well for comparison.

**Illegal Move Oracle (IMO).** Our first attack is based on a very natural idea: the attacker picks the legal move that maximizes the conditional probability of an invalid continuation by the opponent. Formally,

$$f_{IMO}(M, a_1..a_k a_{k+1}) = \underset{a_{k+2} \notin W(a_1..a_{k+1})}{\max} M(a_{k+2}|a_1..a_{k+1}). \tag{2}$$

**Board State Oracle (BSO).** The attacker picks the legal move that maximizes the error of the board state predicted by a given probe $B$ compared to the true board state. This attack is motivated by the hypothesis that the predicted board state has a functional role (a causal effect) on next-token prediction (Nanda et al., 2023; Karvonen, 2024). To be more precise, we maximize the loss of the board state predictor:

$$f_{BSO}(M, a_1..a_k a_{k+1}) = \mathcal{L}_B(M, a_1..a_k a_{k+1}), \tag{3}$$

where $\mathcal{L}_B(M, a_1..a_k a_{k+1})$ is the classification loss of the probe's prediction after the moves $a_1, ..., a_{k+1}$.

**Adversarial Detours (AD) by Vafa et al. (2024).** We include this attack for comparison with related work. Here, the attacker picks the legal move with the lowest conditional probability according to the sequence model:

$$f_{AD}(M, a_1..a_k a_{k+1}) = -M(a_{k+1}|a_1..a_k). \tag{4}$$

Note that this attack is not directed explicitly towards forcing an error; instead, it attempts to guide the generation toward out-of-distribution (OOD) regions.

**Random Move (RM).** As a simple baseline, the adversary randomly selects a legal move in each attack step. That is, $f_{RM}$ is random and independent of its parameters.

**Sequence Model Move (SMM).** The attacker picks the legal move with the highest conditional probability according to the sequence model:

$$f_{SMM}(M, a_1..a_k a_{k+1}) = M(a_{k+1}|a_1...a_k). \tag{5}$$

In practice, this has the effect of simply letting the sequence model generate the sequence, but correcting any incorrect moves by white. That is, the "attacker" is more of a benevolent oracle here.

## 4 OUR SET OF MODELS: ATTEMPTING TO LEARN THE WORLD MODEL

We train a number of models using different training recipes and datasets in order to evaluate the effect of a number of design choices on the quality of the implicit world model.

### 4.1 DATASET CHOICE

The **curated datasets** we used were the following: (1) *MB-500k* with 500k games from the Million-base dataset, consisting of high-quality games, used also by Toshniwal et al. (2022); (2) *Stockfish-8M* with 8M games generated by Karvonen (2024), where the superhuman chess engine Stockfish played as white against engines of varying strength; and (3) *Lichess-16M* with 16M human games obtained from the public Lichess database, also used in Karvonen (2024).

**Random datasets.** Motivated by the findings of Li et al. (2023) and Vafa et al. (2024), who show that models trained on random games learn the true world model better than those that were trained on curated datasets, we use random datasets as well. These contain 500K, 2M, and 10M valid random games, respectively, none of which end due to resignation or agreeing to a draw.

Similar to Toshniwal et al. (2022), we limit the length of every game in the training sets to 150 moves. Longer games are removed from the datasets, and the dataset sizes are given after filtering. For more information about the datasets, please refer to Appendix A.

### 4.2 TRAINING OBJECTIVES

**Tokenization.** We use the tokenizer of Toshniwal et al. (2022), where all squares (e.g. `e2`), and the four possible piece types in promotion (`q`, `r`, `b` and `n`) are represented as single tokens. Thus, all moves are encoded with either 2 or 3 tokens. We also use `BOS` and `EOS` tokens to indicate the start and end of the game, respectively, and a separate `PAD` token for efficient training.

The **next token (NT) prediction objective** aims at predicting the next token after any prefix of any training sequence. While next token prediction is the usual choice, we introduce two additional training objectives that capture certain aspects of the true world model more directly.

The first is the **probability distribution (PD) objective** that aims at capturing all the legal moves simultaneously, as opposed to training only on a single legal target token. This approach is motivated by Vafa et al. (2024), who explicitly define the generated language with the help of the predicted distribution. In the case of our tokenization choice, we need target distributions for move-starting and move-ending tokens. For move-starting tokens, the uniform distribution is used over squares

Table 1: Success rate of each attack strategy over all models. Bold and italic represent the highest and lowest success rates for a model, respectively.

| | Random-500k | | | | Random-2M | | | | Random-10M | | | |
|---|---|---|---|---|---|---|---|---|---|---|---|---|
| | NT | PD | NT+JP | PD+JP | NT | PD | NT+JP | PD+JP | NT | PD | NT+JP | PD+JP |
| RM | 0.954 | 0.975 | 0.954 | 0.984 | 0.854 | 0.931 | 0.848 | 0.930 | 0.673 | 0.874 | 0.699 | 0.839 |
| SMM | *0.419* | 0.881 | *0.493* | *0.810* | *0.408* | 0.943 | *0.476* | 0.954 | *0.172* | 0.900 | *0.192* | 0.845 |
| IMO | **0.996** | **0.999** | **0.996** | **1.000** | **0.999** | **1.000** | **0.997** | **0.999** | **0.972** | **0.992** | **0.976** | **0.988** |
| BSO | 0.886 | *0.875* | 0.816 | 0.872 | 0.779 | *0.858* | 0.745 | *0.901* | 0.541 | *0.811* | 0.528 | *0.757* |
| AD | 0.946 | 0.970 | 0.918 | 0.985 | 0.841 | 0.947 | 0.824 | 0.939 | 0.516 | 0.902 | 0.394 | 0.878 |

| | Millionbase-500k | | | | Stockfish-8M | | | | Lichess-16M | | | |
|---|---|---|---|---|---|---|---|---|---|---|---|---|
| | NT | PD | NT+JP | PD+JP | NT | PD | NT+JP | PD+JP | NT | PD | NT+JP | PD+JP |
| RM | 0.823 | 0.994 | 0.818 | 0.998 | 0.149 | 0.913 | 0.176 | 0.931 | 0.107 | 0.914 | 0.075 | 0.852 |
| SMM | *0.513* | *0.794* | *0.494* | *0.846* | 0.190 | 0.911 | 0.267 | 0.931 | 0.287 | 0.897 | 0.211 | 0.859 |
| IMO | **0.999** | **1.000** | **0.995** | **1.000** | **0.631** | **1.000** | **0.662** | **0.998** | **0.387** | **0.995** | **0.349** | **0.987** |
| BSO | 0.524 | 0.885 | 0.547 | 0.938 | *0.105* | *0.753* | *0.147* | *0.882* | *0.057* | *0.764* | *0.043* | *0.795* |
| AD | 0.806 | 0.989 | 0.803 | 0.994 | 0.142 | 0.893 | 0.150 | 0.915 | 0.066 | 0.898 | 0.067 | 0.883 |

where the player has a movable piece. For move-ending tokens, the target is the uniform distribution over the possible destination squares for the selected piece specified by the previous token. In case of a third promotion token, the uniform distribution is used over the four possible piece type tokens.

The PD objective can also be seen as an explicit way of learning the transition rules of the true world model. This is particularly important for models that are trained on non-random datasets of high-quality chess games, where the next token objective is highly biased by a strategic value function. That is, the next token objective does not allow the model to distinguish between *illegal* and *strategically bad* moves (Li et al., 2023; Vafa et al., 2024).

The second is the **joint probe (+JP) objective**, motivated by recent advances in deep supervision, where training multiple heads on related auxiliary tasks has been used to achieve better performance and consistency (Li et al., 2022a; Zahorodnii, 2025; Huo et al., 2025). We add a linear board state probe to the model, and perform joint training by minimizing the combined loss of the next-token predictor head and the board state probe. This method can be seen as learning to track the world state explicitly.

**Four objectives.** We will use four training objectives, namely standard next-token prediction (NT), probability distribution prediction (PD), next-token prediction combined with board state prediction (NT+JP), and probability distribution prediction with board state prediction (PD+JP).

### 4.3 ARCHITECTURE AND HYPERPARAMETERS

Our models follow the GPT-2 architecture Radford et al. (2019) with 12 hidden layers, 768 hidden dimensions, and 12 attention heads, and a total of 86M parameters. All models were trained for 3 epochs with identical parameters, as detailed in Appendix B.

Every model has an associated **board state probe**. Similar to Vafa et al. (2024), our board state probes take the transformer's last layer representation as input. We only train and evaluate probes on move-ending tokens. If a joint probe was included in the training of a model, we use this probe in our probing experiments. Otherwise, we train a probe for the frozen generative model over 50K games from the model's training set. Further details are presented in Appendix C.

## 5 EXPERIMENTAL RESULTS: ARE IMPLICIT WORLD MODELS SOUND?

Let us first consider the quality of our set of 24 models. Detailed measurements are provided in Appendix D. Here, we highlight that, although models trained on smaller datasets (Random-500k and Millionbase-500k) achieve relatively low legal move ratios between 94.65% and 96.71% on their test sets, the models trained on large datasets (Random-10M, Stockfish-8M, and Lichess-16M)

achieve a ratio between 99.75% and 99.98%, so these models could be considered high-quality if one focused on this metric.

We evaluated every model using our various adversaries. We applied the greedy decoding policy, that is, $m(s) = \arg\max_a M(a|s)$, for all the models. With this policy, having the sequence model play against any non-random adversary results in a deterministic sequence depending only on the starting position. Thus, to evaluate the soundness of a model, we selected 1000 unique prefixes of 10 moves from the training dataset of the model, and performed our adversarial evaluation after each of these warmup sequences, which allowed us to collect statistics and gain fine-grained insights.

The results of the experiments are shown in Table 1 and Figure 1. The table shows the success rate of the 5 adversaries against our 24 models, while the figure also shows the cumulative attack success rate as a function of the number of moves after the warmup sequence.

Clearly, **the implicit world models are not sound.** For most models, at least one adversary achieves close to 100% success rate, indicating severe inconsistencies between the implicit and the true world models. In the following, we make a number of more fine-grained observations based on the results.

### 5.1 ADVERSARIES

**IMO** is always the strongest adversary, usually by a wide margin. Given that IMO directly steers the generative model towards an illegal move, this is not surprising; however, what *is* surprising is that the other two adversaries, namely AD and BSO, are rather weak. This showcases the *need for strong adversaries* in order to reliably verify generative models.

**BSO** shows a mixed performance, but sometimes it is weaker than even the most benign baseline SMM. This implies a *weak causal link* between the correctness of the board state predicted by the probe and the legality of the move predicted by the generative model. We further investigate this phenomenon in Section 6.

**AD** by Vafa et al. (2024) consistently achieves success rates similar to Random Move (RM). An explanation could be that most moves with low conditional probabilities are essentially random from the generative model's perspective. This also shows that a more aggressive attack, such as IMO, is essential for evaluation.

### 5.2 EFFECT OF TRAINING SETUP

**Dataset size matters.** According to Figure 1, it is clear that increasing the dataset size very reliably increases the robustness to our attacks. That is, large datasets increase the level of soundness. This is true independently of dataset type and training objective.

**Random and curated datasets** differ mainly when the next token objective is used for training. With the next token (NT) objective, models trained on curated datasets seem to be very robust, especially when a large dataset is used. At the same time, models trained on random datasets seem to be less robust under the NT objective, compared to the distribution objective PD, sometimes significantly so (see also Appendix E on this topic). However, in Section 7 we demonstrate that when executing the attacks using an out-of-distribution warmup sequence, the curated models are much less sound. We discuss the possible reasons in Section 7.

**Multi-task learning does not help.** Adding a joint probe to the training scheme has a negligible effect on the soundness of the implicit world model. We also investigate this in Section 6.

**Models overfit sequence length.** Figure 1 also reveals that many models, especially those trained on large datasets with the PD objective, *strongly overfit the sequence length*. This is evident from the fact that after exceeding the sequence lengths available in the dataset (up to 150), the models suddenly become extremely unreliable. This alarming finding suggests that the models do not use abstract board state representations internally that would be independent of sequence length.

## 6 ON THE CAUSALITY OF BOARD STATE PROBES

Here, we investigate the connection between the board state probes and the next-token predictor heads. As observed in Section 5, attacking the board state probe is not an effective strategy, and

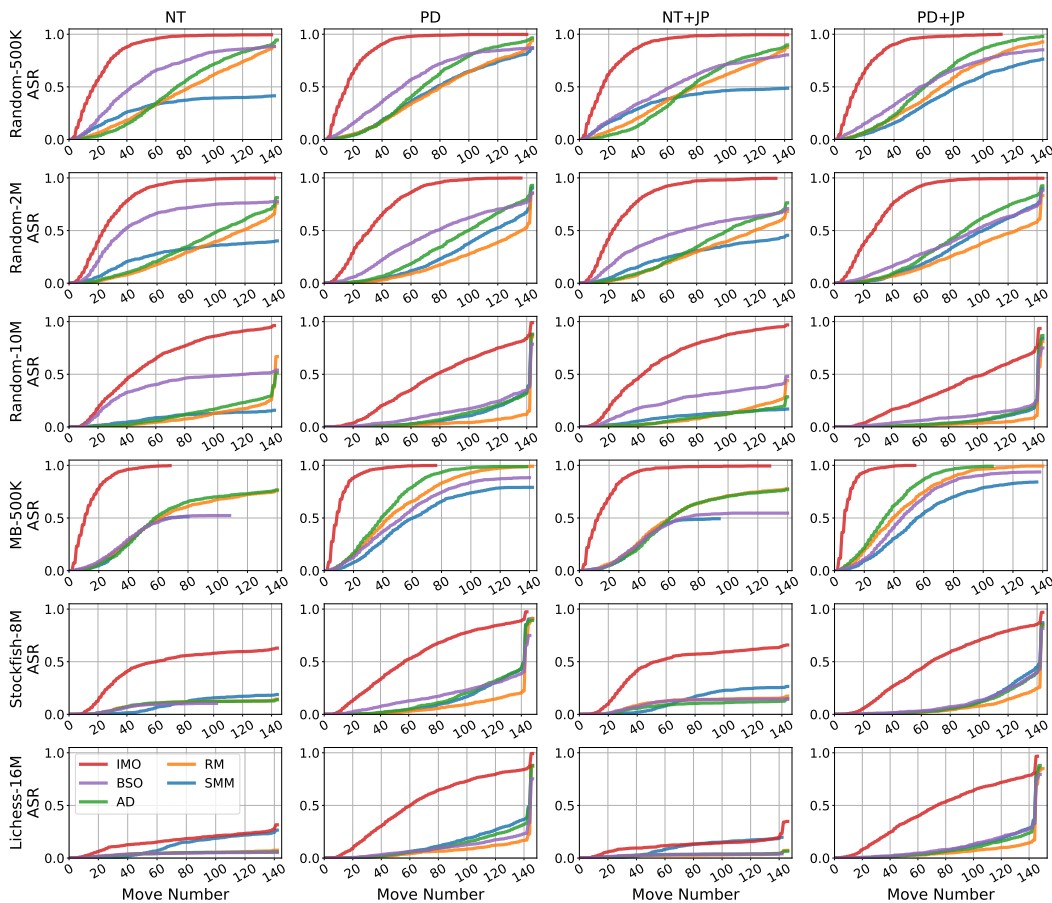

Figure 1: Attack dynamics demonstrated by the move-wise attack success rate (ASR) for each dataset (row) and model (column). On each plot, the X-axis shows the move number, and the Y-axis shows the ASR attained by the attacks. Stronger attacks increase ASR more quickly. All lines stop at the move when the attack reached its final ASR reported in Table 1.

Table 2: Mean sample-wise cosine distance of the gradients of the next-token head and the board state probe w.r.t. their input embeddings.

| | NT | PD | NT+JP | PD+JP |
|---|---|---|---|---|
| R-500K | 0.989 | 0.986 | 0.982 | 0.979 |
| R-2M | 0.990 | 0.989 | 0.988 | 0.992 |
| R-10M | 0.989 | 0.989 | 0.992 | 1.000 |
| MB-500K | 0.984 | 0.982 | 0.975 | 0.975 |
| SF-8M | 0.978 | 0.989 | 0.989 | 1.000 |
| LC-16M | 0.966 | 0.988 | 0.990 | 1.000 |

Table 3: Ratio of illegal moves under the BSO attack where the predicted illegal move is legal in the board state obtained via probing.

| | NT | PD | NT+JP | PD+JP |
|---|---|---|---|---|
| R-500K | 0.386 | 0.477 | 0.365 | 0.460 |
| R-2M | 0.279 | 0.328 | 0.107 | 0.259 |
| R-10M | 0.144 | 0.105 | 0.051 | 0.044 |
| MB-500K | 0.305 | 0.514 | 0.258 | 0.575 |
| SF-8M | 0.181 | 0.124 | 0.034 | 0.184 |
| LC-16M | 0.193 | 0.093 | 0.023 | 0.142 |

multi-task training with a board state probe (+JP) achieves negligible improvements in soundness. The former observation suggests a weak causal link between the obtained board state and the prediction of the model, and the latter one provides additional evidence that the probe functions independently of the next-token predictor head.

**Representation gradients.** We investigate these hypotheses further using gradient-based alignment analysis. Let us assume that $x(s)$ is the last token of the final-layer representation of $M$ over an action sequence $s$. In our setup, $x(s)$ is also the input of both the next-token predictor head and the board state probe. We will consider the gradient of the loss terms according to $x(s)$, namely

$g_B = \nabla_{x(s)} \mathcal{L}_B(M, s)$ and $g_{NT} = \nabla_{x(s)} \mathcal{L}_{NT}(M, s)$. Depending on the model in question, $\mathcal{L}_{NT}$ is either the hard next-token loss or the soft PD loss.

**The heads are independent.** We calculate the average cosine distance between $g_{NT}$ and $g_B$ over 10,000 games from each model's training set and present them in Table 2. In all the cases—including the joint probe objectives—the gradient of the board state probe is almost orthogonal to that of the next-token head, which indicates that the two tasks rely on independent subspaces of the representation. This finding is the exact opposite of the hypothesis that motivated the use of a joint probe, namely that training a board state probe will encourage a better representation of the board state, thereby increasing the soundness of the implicit world model as well.

**BSO attack success is mostly independent of probe.** Table 3 shows the ratio of those illegal moves enforced by the BSO attack that are also illegal according to the board state probe. Especially for large datasets, this ratio is very low, indicating that even when the BSO attack is successful, it is *not* due to misleading the board state predictor. This indicates that the probe is more aligned with the ground truth than the model's prediction, further supporting a limited causal link between the predicted board state and the model's prediction.

## 7 ARE SEEMINGLY SOUND MODELS REALLY SOUND?

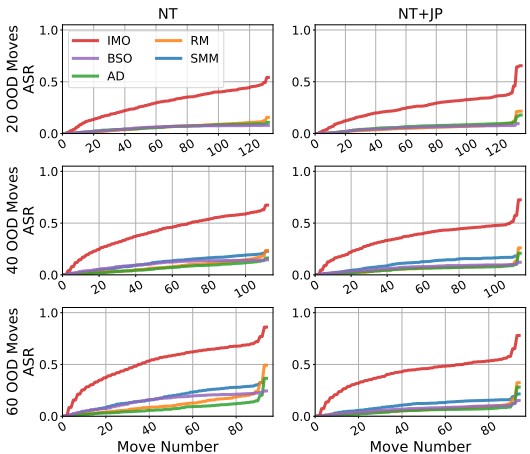

Figure 2: Attack dynamics demonstrated similarly to Figure 1 for NT (top row) and NT+JP (bottom row) models trained on the Lichess-16M dataset, evaluated with out-of-distribution (OOD) warmup sequences of varying lengths.

Our results in Section 5 indicated a surprising level of soundness for the models trained on the larger curated datasets, and especially on Lichess-16M with next-token prediction (NT). Here, we argue that these models are not actually sound. We hypothesize that the apparent soundness of models trained on large curated datasets has to do with a strong gravitation to in-distribution trajectories. This, in turn, is most likely due to predicting not only legal, but also strategically good moves, resulting in a much more focused distribution that assigns a high probability to far fewer moves than other models.

To test this hypothesis, we applied random but valid warmup sequences that are not in the training set. We evaluated 1000 such out-of-distribution warmup sequences of different lengths: 10, 20, and 30 moves per player. Table 4 shows the success rates of each attack, along with the difference from the original evaluation with in-distribution warmup sequences, and Figure 2 shows the corresponding attack dynamics.

The attack success rate significantly increases as we make the initial board state more and more out-of-distribution. This indicates that the model does not capture the true abstract transition rules.

## 8 ON THE IMPACT OF DECODING POLICY

In this section, we investigate the impact that different, sampling-based decoding policies have on our analysis framework. Here, we focus on top-$k$ sampling (Fan et al., 2018) and we further investigate top-$p$ sampling (Holtzman et al., 2020) in Appendix E, where we observed highly similar results.

**Experimental setup.** We use top-$k$ sampling with $k = 4$. In order to remain consistent with our earlier experiments, we used the same 1000 warmup sequences in our evaluations. Since the move sequence is now non-deterministic, we perform three sets of evaluations with different random seeds and report the average ASR achieved by our attacks over these evaluations.

Table 4: Success rate of each attack strategy over the NT and NT+JP models trained on the Lichess-16M dataset, when evaluated with out-of-distribution (OOD) warmup prefixes of varying lengths. The increases in ASR compared to evaluations with in-distribution prefixes (as seen in Table 1) are in brackets.

|  | 20 OOD Moves | | 40 OOD Moves | | 60 OOD Moves | |
|---|---|---|---|---|---|---|
|  | NT | NT+JP | NT | NT+JP | NT | NT+JP |
| RM | 0.244 (+0.14) | 0.221 (+0.15) | 0.360 (+0.25) | 0.272 (+0.20) | 0.495 (+0.39) | 0.343 (+0.27) |
| SMM | 0.203 (-0.08) | 0.145 (-0.07) | 0.244 (-0.04) | 0.200 (-0.01) | 0.351 (+0.06) | 0.217 (+0.01) |
| IMO | 0.667 (+0.28) | 0.659 (+0.31) | 0.785 (+0.40) | 0.735 (+0.39) | 0.864 (+0.48) | 0.797 (+0.45) |
| BSO | 0.080 (+0.02) | 0.097 (+0.05) | 0.148 (+0.09) | 0.124 (+0.08) | 0.245 (+0.19) | 0.154 (+0.11) |
| AD | 0.165 (+0.10) | 0.183 (+0.12) | 0.270 (+0.20) | 0.225 (+0.16) | 0.374 (+0.31) | 0.294 (+0.23) |

Table 5: Success rate of each attack strategy over all models with the top-$k$ decoding strategy ($k = 4$). Results are averaged over three separate evaluations over the same set of warmup sequences. Bold and italic represent the highest and lowest success rates for a model, respectively.

|  | Random-500k | | | | Random-2M | | | | Random-10M | | | |
|---|---|---|---|---|---|---|---|---|---|---|---|---|
|  | NT | PD | NT+JP | PD+JP | NT | PD | NT+JP | PD+JP | NT | PD | NT+JP | PD+JP |
| RM | 0.963 | 0.990 | 0.969 | 0.993 | 0.886 | 0.971 | 0.902 | 0.976 | 0.703 | 0.903 | 0.750 | 0.908 |
| SMM | *0.937* | 0.994 | *0.937* | 0.997 | *0.745* | 0.973 | *0.819* | 0.984 | *0.315* | 0.913 | *0.386* | 0.926 |
| IMO | **0.998** | **1.000** | **0.999** | **0.999** | **0.995** | **0.997** | **0.998** | **0.999** | **0.958** | **0.974** | **0.959** | **0.959** |
| BSO | 0.961 | *0.979* | 0.960 | *0.978* | 0.895 | *0.954* | 0.903 | *0.968* | 0.706 | *0.885* | 0.750 | *0.884* |
| AD | 0.982 | 0.991 | 0.985 | 0.993 | 0.969 | 0.975 | 0.977 | 0.979 | 0.944 | 0.938 | 0.955 | 0.919 |

|  | Millionbase-500k | | | | Stockfish-8M | | | | Lichess-16M | | | |
|---|---|---|---|---|---|---|---|---|---|---|---|---|
|  | NT | PD | NT+JP | PD+JP | NT | PD | NT+JP | PD+JP | NT | PD | NT+JP | PD+JP |
| RM | 0.954 | 0.998 | 0.951 | 0.996 | 0.326 | 0.926 | 0.365 | 0.930 | 0.246 | 0.912 | 0.189 | 0.922 |
| SMM | 0.952 | 0.999 | 0.968 | **1.000** | *0.263* | 0.943 | *0.281* | 0.949 | 0.623 | 0.910 | 0.607 | 0.922 |
| IMO | **0.998** | **1.000** | **0.996** | **1.000** | **0.787** | **0.981** | **0.815** | **0.985** | **0.654** | **0.979** | **0.616** | **0.977** |
| BSO | *0.938* | *0.993* | *0.927* | *0.990* | 0.366 | *0.909* | 0.394 | *0.913* | 0.240 | *0.871* | 0.194 | *0.902* |
| AD | 0.943 | 0.996 | 0.936 | 0.998 | 0.322 | 0.935 | 0.344 | 0.943 | *0.192* | 0.932 | *0.156* | 0.933 |

**Results.** Table 5 shows the average ASR of our attacks when our models use the top-$k$ decoding policy. All attacks achieve a higher ASR compared to the results against the greedy decoding policy, but otherwise their relative performance is similar, showcasing that our results are robust to the decoding policy used to generate sequences. This is particularly interesting in the case of the IMO attack, which assumes a greedy decoding policy. These results imply that steering the model towards states that maximize the probability of the top-1 error will also maximize the overall probability of error, suggesting that IMO is able to uncover vulnerabe state-regions, that is, gaps in the model's knowledge as opposed to just one-off errors.

## 9    CONCLUSIONS AND LIMITATIONS

We proposed adversarial sequence generation to test the soundness of implicit world models. The most successful attack was IMO based on an explicit lookahead search for illegal moves. Our methodology allowed us not only to prove that none of the training setups resulted in sound models, but also to observe interesting patterns, such as the importance of using a large dataset, the misleading appearance of soundness in the case of a high-quality, large gameplay dataset, and the positive effect of using a probability distribution objective. At the same time, we found that board state probes do not help much in any form we tried, and seem to be mostly independent of generation.

Our main limitation is that, similar to other seminal works in the field (Vafa et al., 2024; Li et al., 2023), we rely on one generative sequence model architecture due to the expensive training and evaluation. Although this study provides compelling arguments behind our proposed methodology, the effect of different architectures would certainly be interesting to analyse in the future.

## 10 ACKNOWLEDGEMENTS

This work was supported by the University Research Grant Program of the Ministry for Culture and Innovation from the source of the National Research, Development and Innovation Fund, and by project 2024-1.2.3-HU-RIZONT-2024-00017, implemented with the support provided by the Ministry of Culture and Innovation of Hungary from the National Research, Development and Innovation Fund, financed under the 2024-1.2.3-HU-RIZONT funding scheme.

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

Table 6: Number of tokens and moves for each dataset, along with the average game lengths (standard deviation indicated in brackets).

| Dataset | Number of Tokens | Number of Moves |
|---|---|---|
| Random-500K | 97,974,729 | 48,389,735 |
| | Avg. per game: 195.95 (+/-72.20) | Avg. per game: 96.78 (+/-35.97) |
| Random-2M | 392,296,887 | 193,756,892 |
| | Avg. per game: 196.15 (+/-72.06) | Avg. per game: 96.88 (+/-35.89) |
| Random-10M | 1,962,204,706 | 969,142,950 |
| | Avg. per game: 196.22 (+/-71.93) | Avg. per game: 96.91 (+/-35.84) |
| Millionbase-500K | 75,964,911 | 37,469,929 |
| | Avg. per game: 151.93 (+/-57.52) | Avg. per game: 74.94 (+/-28.73) |
| Stockfish-8M | 1,302,918,935 | 641,501,357 |
| | Avg. per game: 163.97 (+/-66.97) | Avg. per game: 80.73 (+/-33.29) |
| Lichess-16M | 2,311,925,519 | 1,138,275,036 |
| | Avg. per game: 142.57 (+/-53.74) | Avg. per game: 70.19 (+/-26.75) |

## A  DATASET DETAILS

In our evaluations, we used three randomly generated datasets and three curated datasets. All datasets contain only legal game sequences. We rounded the sizes of the Stockfish-8M and Lichess-16M datasets (Karvonen, 2024), as they contain 7,946,149 and 16,216,625 games after filtering, respectively. The number of moves and tokens in each dataset is shown in Table 6, and the distribution of game lengths is shown in Figure 3.

All games in the random datasets, as well as the StockFish-8M dataset, end according to the rules (i.e., by checkmate, stalemate, draw by repetition, or draw by insufficient material). However, human games in the Millionbase-500K and Lichess-16M datasets can end prematurely (i.e., by one player resigning, both players agreeing on a draw, or, in rare cases, a player running out of time). In the tokenized game sequences, this phenomenon shows up as the EOF token – which is always used to indicate the end of the game – being at the end of a sequence where the game is not over according to the rules.

While 70.22% of games in the Lichess-16M dataset, and a staggering 94.37% of games in the Millionbase-500K dataset, end prematurely, usually immediately after a player makes a strategic blunder, we found this to have little effect on the soundness of the implicit world models. We detail these findings in Appendix E.

## B  MODEL TRAINING DETAILS

We used the GPT-2 implementation of the `transformers`[2] library (Wolf et al., 2019). Our hyper-parameter setting closely follows that of Toshniwal et al. (2022). All our models were trained for 3 epochs using the AdamW optimizer (Loshchilov & Hutter, 2019), with a learning rate of $3 \times 10^{-4}$, and an $L_2$ weight decay of 0.01. The learning rate is warmed up linearly over the first 10% of training, followed by a linear decay. We used a batch size of 128 and accumulated gradients over 4 batches before each optimizer step. We did not use mixed-precision training. Depending on the dataset size, training a model took between 70 minutes and 37 hours on a single Nvidia H100 GPU.

For the joint probe (+JP) training objective, we experimented with various scaling factors for the loss of the board state probe in our initial exploration phase, but found no meaningful difference between

---

[2]Specifically, version 4.55.3, as compatibility with other versions is not guaranteed.

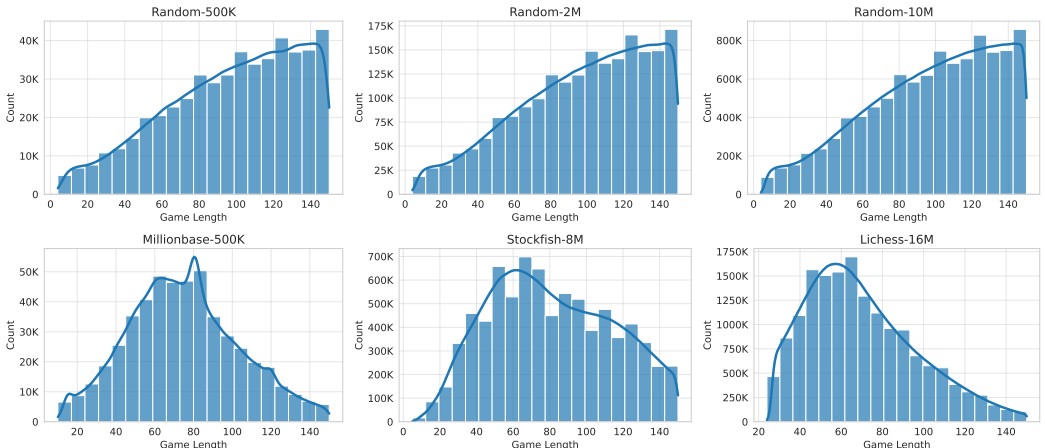

Figure 3: Distribution of game lengths in our datasets.

different settings. We don't apply any scaling to any loss term and note that the joint probe's loss is typically a fifth of the next token predictor's loss.

### B.1 ILLUSTRATING THE PROBABILITY DISTRIBUTION OBJECTIVE

Figure 4 illustrates the probability distribution (PD) objective for the first three moves of a game. After a game prefix, the model is trained to learn the probability distribution of valid single-token continuations.

## C PROBE TRAINING DETAILS

Our linear board state probes are trained to predict the board state at the end of a move sequence from the final-layer representation of the language model. We only train and evaluate probes on move-ending tokens, i.e., for moves comprised of two tokens (e.g. e2e4), we use the representation of the destination square token, and for moves comprised of three tokens (e.g. e7e8q), the promotion piece type token's representation is used. This is motivated by the fact that only after processing the last token of the move should the move be completed in the language model's internal model. We rely on a separate oracle to know which tokens are move-ending, not the language model itself.

In formulating the targets for the board state classification problem, we use an absolute encoding just like Li et al. (2023), where a piece's label is always the same, regardless of which player's turn it is. In contrast, Karvonen (2024) and Nanda et al. (2023) use a side-specific encoding, where the labels of the pieces depend on which player is to move. Nanda et al. (2023) show that absolute encoding is harder for probes to learn, but our probes achieve comparable (and in some cases superior) accuracies to those in Karvonen (2024), as showcased in Section D.

When probes are not jointly trained with the language model, we train them after the model is trained and frozen. Our training parameters are inspired by Karvonen (2024). We train our probes on 50,000 games from the model's own training set for 1 epoch using the AdamW optimizer with betas (0.9, 0.99), an initial learning rate of $10^{-3}$ and $L_2$ weight decay of 0.01. The batch size, i.e., the number of move-ending token representations per optimization step, was 4096, and we decayed the learning rate to $10^{-4}$ after 1000 optimization steps.

## D PERFORMANCE METRICS

Table 7 shows the perplexities of our models, evaluated over 15,000 test games that were unseen by either the model or the probe during training. While perplexity does not measure the soundness of the implicit world model, the values show that the joint probe (+JP) objective fails to achieve meaningful (or, in some cases, any) improvement in the model's performance.

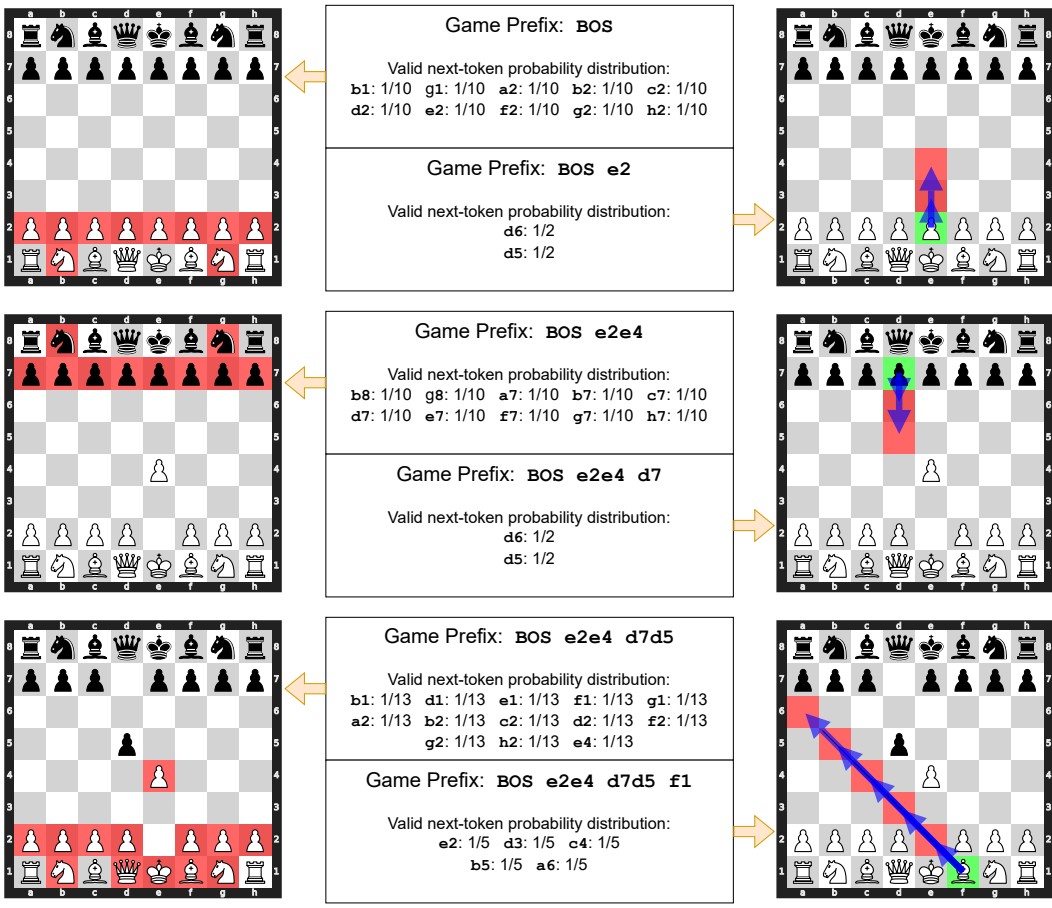

Figure 4: Illustration of the probability distribution (PD) objective using the first three moves of a game. Boards on the left highlight movable pieces after a sequence of completed moves, indicating the possible move-starting squares. A uniform probability distribution is assigned to the tokens corresponding to these squares. Boards on the right highlight the possible destination squares in red, once a starting square (highlighted with green) is available. A uniform probability distribution is assigned to these possible move-ending squares as well.

Table 7: Model perplexities. We report the standard, token-wise perplexity, as opposed to canonical (move-wise) perplexity reported by Toshniwal et al. (2022).

|  | R500k | R2M | R10M | MB500K | SF8M | LC8M |
|---|---|---|---|---|---|---|
| NT | 5.9478 | 5.7139 | 5.7574 | 3.1347 | 2.3756 | 2.1577 |
| PD | 6.5096 | 6.1893 | 6.2446 | 6.6601 | 5.0957 | 5.9161 |
| NT+JP | 6.0369 | 5.7428 | 5.7812 | 3.1480 | 2.3820 | 2.1581 |
| PD+JP | 6.7879 | 6.2740 | 6.3030 | 6.9640 | 5.0999 | 5.9257 |

The perplexities of models trained with the probability distribution (PD) objective are naturally lower, as the model is trained not to assign a high probability to the actual next token in the sequence but to approximate the probability distribution of valid single-token continuations. As a result, the model's confidence for the actual next token will be lower, which in turn increases perplexity.

Table 8 shows the ratio of legal moves played by our models in 10,000 games that were unseen by each model during training. While models trained on smaller datasets (Random-500k and Millionbase-500k) achieve relatively low legal move ratios between 94.65% and 96.71%, models

Table 8: Ratio of legal moves of our models on 10,000 test games that were unseen by the models during training.

|  | R500k | R2M | R10M | MB500K | SF8M | LC8M |
|---|---|---|---|---|---|---|
| NT | 0.9634 | 0.9914 | 0.9986 | 0.9640 | 0.9977 | 0.9985 |
| PD | 0.9539 | 0.9862 | 0.9985 | 0.9671 | 0.9991 | 0.9998 |
| NT+JP | 0.9612 | 0.9883 | 0.9985 | 0.9638 | 0.9975 | 0.9983 |
| PD+JP | 0.9465 | 0.9772 | 0.9989 | 0.9597 | 0.9990 | 0.9997 |

trained on large datasets (Random-10M, Stockfish-8M, and Lichess-16M) achieve high legal move ratios between 99.75% and 99.98%.

However, as argued by Vafa et al. (2024) and demonstrated by our results, legality ratio is only a surface-level metric and does not reflect on the soundness of the implicit world model.

Tables 9 and 10 show the move-wise mean accuracies and piece accuracies of our board state probes, evaluated over 15,000 test games that were unseen by either the model or the probe during training. Piece accuracy is defined as accuracy over squares that either contain pieces (i.e., they are not empty) or are predicted by the probe to contain pieces.

While most probes achieve remarkably high accuracies (on par with, or even higher than, the probing accuracy reported in Karvonen (2024)), it must be noted that probes, especially those that were not jointly trained with the model, are biased towards empty squares. As shown in Figure 5, towards the later parts of the game, probes get progressively worse at predicting pieces on the board, but their accuracies stay high due to the large number of empty squares that the probe is able to correctly guess.

To correct this bias, we experimented with weighting the loss term of the board state probe per square, based on whether it is a "piece square" (i.e., a square that either contains a piece or is predicted by the probe to contain a piece) or an empty square. Our goal was to apply an increased weight to piece squares, thereby forcing the probe to learn to track the pieces better. We applied weights between 2 and 20 to piece squares in preliminary experiments, the results of which showed minor improvements in piece accuracy at a minor cost of overall accuracy, but these probes showed no difference compared to the standard probes when used as the basis of the BSO adversary in our framework.

While we believe this bias towards empty squares represents a fundamental issue, its relevance to our findings is minimal, especially in light of the aforementioned weighting experiments. We leave it up to future work to create linear probe training methods that properly address this challenge.

Table 9: Move-wise average accuracies of our board state probes.

|  | R500k | R2M | R10M | MB500K | SF8M | LC8M |
|---|---|---|---|---|---|---|
| NT | 0.8416 | 0.9178 | 0.9554 | 0.9014 | 0.9640 | 0.9698 |
| PD | 0.8237 | 0.8754 | 0.9410 | 0.8849 | 0.9584 | 0.9732 |
| NT+JP | 0.9831 | 0.9996 | 1.0000 | 0.9879 | 0.9999 | 1.0000 |
| PD+JP | 0.9786 | 0.9982 | 1.0000 | 0.9851 | 1.0000 | 1.0000 |

# E    FURTHER EXPERIMENTAL RESULTS

In this section, we present further experimental results supporting our claims.

Table 10: Move-wise average piece accuracies of our board state probes.

|       | R500k  | R2M    | R10M   | MB500K | SF8M   | LC8M   |
|-------|--------|--------|--------|--------|--------|--------|
| NT    | 0.6005 | 0.7856 | 0.8812 | 0.7473 | 0.8858 | 0.9182 |
| PD    | 0.5634 | 0.6840 | 0.8445 | 0.7041 | 0.8671 | 0.9264 |
| NT+JP | 0.9546 | 0.9989 | 1.0000 | 0.9675 | 0.9998 | 0.9999 |
| PD+JP | 0.9427 | 0.9951 | 1.0000 | 0.9600 | 1.0000 | 1.0000 |

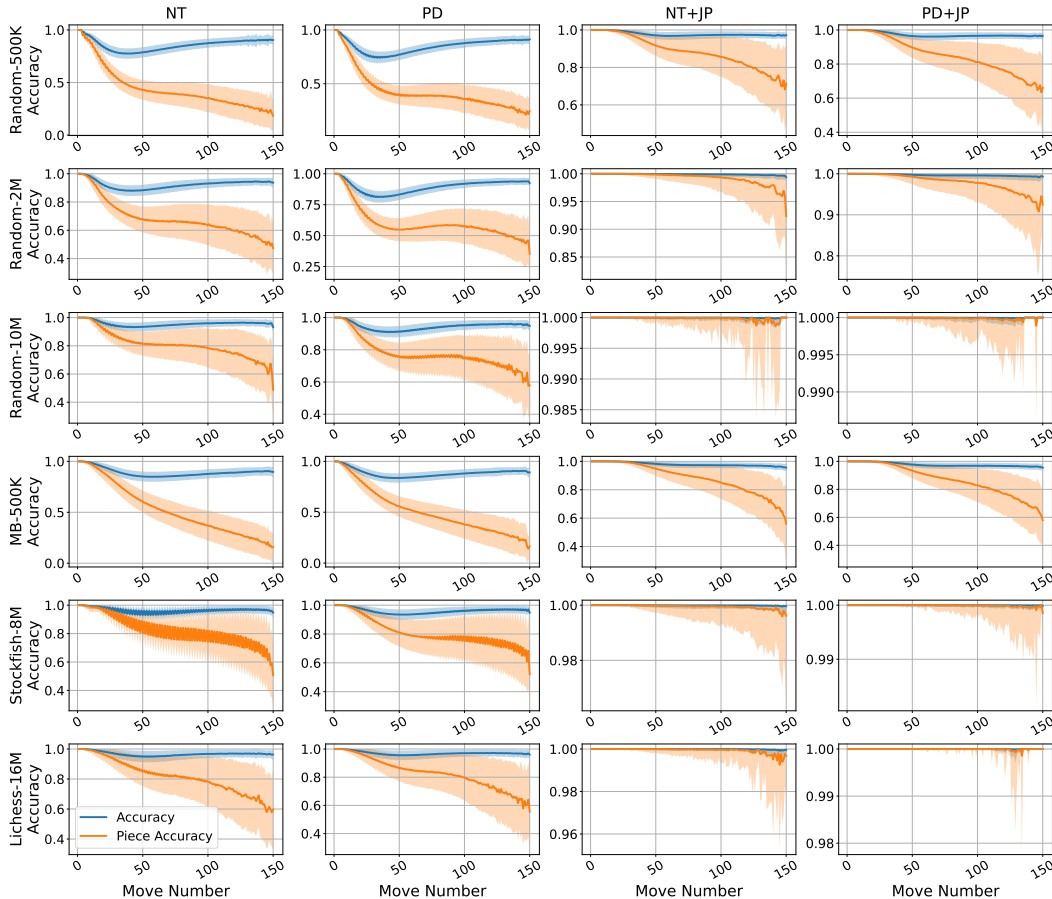

Figure 5: Move-wise mean board state probe accuracies and piece accuracies. Error bars represent one standard deviation.

## E.1 MODEL RESILIENCE TO OUR ADVERSARIES

Table 11 shows the average lengths of the sequences generated by the various adversaries playing against all models, without counting the length of the warmup sequence, regardless of whether the adversary succeeds. These results complement Table 1, as stronger attacks yield shorter sequences, while weaker attacks result in longer sequences. From a different point of view, longer sequences for the same adversary show an increase in resilience by the models.

Interestingly, models trained with the probability distribution (PD) objective are harder to attack than regular next-token (NT) models. This is especially true for weaker adversaries, where PD can achieve a nearly $3\times$ increase in sequence lengths. This supports the notion that PD is a more explicit way of learning the rules, while NT models learn inconsistent and possibly fragmented rules. On the

Table 11: Average sequence length under adversarial conditions.

| | Random-500k | | | | Random-2M | | | | Random-10M | | | |
|---|---|---|---|---|---|---|---|---|---|---|---|---|
| | NT | PD | NT+JP | PD+JP | NT | PD | NT+JP | PD+JP | NT | PD | NT+JP | PD+JP |
| RM | 79.0 | 80.6 | 77.8 | 72.3 | 92.9 | 112.2 | 92.1 | 105.2 | 106.5 | 131.3 | 106.9 | 131.8 |
| SMM | 45.3 | 78.1 | 47.8 | 78.1 | 52.6 | 105.0 | 61.0 | 96.5 | 46.7 | 125.5 | 48.6 | 128.4 |
| IMO | 20.9 | 18.7 | 20.1 | 19.0 | 27.9 | 34.6 | 27.9 | 29.7 | 51.5 | 81.4 | 50.1 | 95.4 |
| BSO | 46.1 | 54.9 | 57.3 | 57.7 | 35.4 | 71.9 | 57.6 | 85.2 | 43.9 | 113.2 | 67.4 | 116.6 |
| AD | 73.0 | 70.2 | 72.1 | 62.4 | 84.5 | 93.2 | 83.4 | 86.7 | 96.7 | 124.3 | 91.6 | 129.1 |
| | Millionbase-500k | | | | Stockfish-8M | | | | Lichess-16M | | | |
| | NT | PD | NT+JP | PD+JP | NT | PD | NT+JP | PD+JP | NT | PD | NT+JP | PD+JP |
| RM | 65.4 | 49.4 | 62.7 | 43.7 | 51.8 | 129.1 | 53.9 | 129.9 | 50.4 | 131.7 | 52.6 | 130.8 |
| SMM | 48.8 | 59.1 | 50.7 | 57.3 | 72.4 | 120.6 | 73.5 | 124.5 | 84.8 | 120.4 | 82.9 | 125.4 |
| IMO | 13.5 | 10.8 | 16.0 | 8.4 | 49.0 | 65.9 | 51.7 | 74.5 | 65.3 | 70.2 | 70.7 | 80.0 |
| BSO | 48.7 | 51.5 | 53.2 | 47.5 | 46.7 | 103.6 | 48.9 | 120.7 | 41.4 | 117.7 | 44.9 | 119.4 |
| AD | 61.6 | 40.9 | 62.1 | 35.9 | 46.5 | 118.6 | 52.6 | 125.0 | 45.4 | 123.9 | 47.0 | 128.3 |

Table 12: Ratio of games that end in either checkmate or stalemate where the model correctly identifies the end of the game by predicting the `EOS` token after the final move and nowhere else.

| | R500k | R2M | R10M | MB500K | SF8M | LC8M |
|---|---|---|---|---|---|---|
| NT | 0.1865 | 0.3330 | 0.6806 | 0.0388 | 0.2510 | 0.3122 |
| PD | 0.1686 | 0.2948 | 0.6955 | 0.0000 | 0.2744 | 0.6139 |
| NT+JP | 0.1846 | 0.3236 | 0.6503 | 0.0397 | 0.2413 | 0.3280 |
| PD+JP | 0.1533 | 0.2528 | 0.7617 | 0.0000 | 0.2926 | 0.5635 |

other hand, the joint probe (+JP) objective has minimal impact on the models' resilience, furthering our claim that the board state probe is largely independent of the next-token predictor head.

### E.2 THE IMPACT OF PREMATURELY ENDED GAMES

As mentioned in Appendix A, our two datasets of human games contain a high ratio of games that end prematurely. Here, we investigate if this has any effect on the models and the adversarial evaluation.

We evaluate the models' ability to correctly predict the end of the game, on 10000 games that end in checkmate and 1000 games that end in stalemate. All games were unseen by all models. We say a model is able to accurately identify the end of the game if, when processing the entire sequence, it predicts the `EOS` token after the final move, and nowhere before.

Table 12 shows the accuracies of all our models in predicting the end of the game. It is clear that the nature of the dataset (random or curated) has more impact on the models' ability to identify the end of the game than the ratio of prematurely ended games. Models trained on the Stockfish-8M dataset, a dataset without prematurely ended games, still perform poorly, while models trained on the largest random dataset (which is only slightly larger than Stockfish-8M) are significantly better at predicting the end of the game.

However, it is still possible that the mistake the adversaries force the models to make is incorrectly predicting the end of the game. One could assume that, for models whose training data has a very high ratio of prematurely ended games, this type of error would dominate the adversarial evaluation. While this would not mean the implicit world models are sound, a phenomenon like this would still cast shade on our results by suggesting that we simply identified overfitting in our models.

Table 13: The success rate of our attacks against our models broken down into its two possible sources of success: forcing the model to predict an illegal move (top half), and forcing the model to incorrectly predict the end of the game (bottom half).

### Attack Success Rate due to **Illegal Move**

| | Random-500k | | | | Random-2M | | | | Random-10M | | | |
|---|---|---|---|---|---|---|---|---|---|---|---|---|
| | NT | PD | NT+JP | PD+JP | NT | PD | NT+JP | PD+JP | NT | PD | NT+JP | PD+JP |
| RM | 0.609 | 0.762 | 0.612 | 0.831 | 0.298 | 0.346 | 0.311 | 0.511 | 0.093 | 0.062 | 0.106 | 0.047 |
| SMM | 0.272 | 0.769 | 0.349 | 0.731 | 0.188 | 0.483 | 0.242 | 0.660 | 0.060 | 0.166 | 0.054 | 0.107 |
| IMO | 0.842 | 0.855 | 0.851 | 0.868 | 0.695 | 0.821 | 0.752 | 0.878 | 0.445 | 0.507 | 0.452 | 0.361 |
| BSO | 0.700 | 0.763 | 0.660 | 0.791 | 0.385 | 0.498 | 0.502 | 0.655 | 0.134 | 0.148 | 0.145 | 0.119 |
| AD | 0.803 | 0.845 | 0.813 | 0.891 | 0.611 | 0.604 | 0.612 | 0.726 | 0.256 | 0.184 | 0.157 | 0.134 |

| | Millionbase-500k | | | | Stockfish-8M | | | | Lichess-16M | | | |
|---|---|---|---|---|---|---|---|---|---|---|---|---|
| | NT | PD | NT+JP | PD+JP | NT | PD | NT+JP | PD+JP | NT | PD | NT+JP | PD+JP |
| RM | 0.724 | 0.989 | 0.766 | 0.993 | 0.121 | 0.178 | 0.147 | 0.197 | 0.049 | 0.151 | 0.029 | 0.121 |
| SMM | 0.484 | 0.791 | 0.478 | 0.843 | 0.045 | 0.382 | 0.054 | 0.407 | 0.030 | 0.341 | 0.030 | 0.247 |
| IMO | 0.995 | 0.998 | 0.990 | 1.000 | 0.552 | 0.701 | 0.561 | 0.610 | 0.173 | 0.724 | 0.155 | 0.683 |
| BSO | 0.512 | 0.883 | 0.544 | 0.938 | 0.079 | 0.231 | 0.129 | 0.349 | 0.042 | 0.176 | 0.037 | 0.257 |
| AD | 0.732 | 0.989 | 0.758 | 0.994 | 0.121 | 0.330 | 0.114 | 0.319 | 0.049 | 0.300 | 0.041 | 0.215 |

### Attack Success Rate due to **Incorrectly Predicted Game Ending**

| | Random-500k | | | | Random-2M | | | | Random-10M | | | |
|---|---|---|---|---|---|---|---|---|---|---|---|---|
| | NT | PD | NT+JP | PD+JP | NT | PD | NT+JP | PD+JP | NT | PD | NT+JP | PD+JP |
| RM | 0.251 | 0.092 | 0.237 | 0.096 | 0.356 | 0.180 | 0.307 | 0.067 | 0.164 | 0.056 | 0.166 | 0.037 |
| SMM | 0.141 | 0.041 | 0.136 | 0.032 | 0.208 | 0.190 | 0.195 | 0.084 | 0.092 | 0.163 | 0.112 | 0.071 |
| IMO | 0.153 | 0.143 | 0.144 | 0.132 | 0.303 | 0.179 | 0.245 | 0.119 | 0.502 | 0.331 | 0.501 | 0.351 |
| BSO | 0.181 | 0.104 | 0.139 | 0.061 | 0.390 | 0.269 | 0.180 | 0.076 | 0.382 | 0.199 | 0.267 | 0.097 |
| AD | 0.094 | 0.090 | 0.065 | 0.089 | 0.120 | 0.189 | 0.088 | 0.106 | 0.037 | 0.132 | 0.040 | 0.063 |

| | Millionbase-500k | | | | Stockfish-8M | | | | Lichess-16M | | | |
|---|---|---|---|---|---|---|---|---|---|---|---|---|
| | NT | PD | NT+JP | PD+JP | NT | PD | NT+JP | PD+JP | NT | PD | NT+JP | PD+JP |
| RM | 0.035 | 0.000 | 0.013 | 0.000 | 0.011 | 0.023 | 0.009 | 0.047 | 0.024 | 0.016 | 0.010 | 0.021 |
| SMM | 0.029 | 0.000 | 0.016 | 0.000 | 0.136 | 0.047 | 0.201 | 0.037 | 0.212 | 0.024 | 0.155 | 0.035 |
| IMO | 0.004 | 0.002 | 0.005 | 0.000 | 0.065 | 0.186 | 0.082 | 0.243 | 0.101 | 0.116 | 0.038 | 0.101 |
| BSO | 0.012 | 0.000 | 0.002 | 0.000 | 0.026 | 0.152 | 0.016 | 0.044 | 0.012 | 0.053 | 0.001 | 0.016 |
| AD | 0.035 | 0.000 | 0.012 | 0.000 | 0.006 | 0.101 | 0.010 | 0.074 | 0.010 | 0.017 | 0.003 | 0.019 |

Table 13 breaks down the attack success rate (ASR) achieved by every adversary against our models into two components: ASR due to forcing the model to predict an illegal move, and ASR due to forcing the model to incorrectly predict the end of the game. In almost all cases, the vast majority of successful attacks force the model to predict illegal moves, even when the models were trained on datasets that contain many prematurely ended games. Among the few exceptions, the IMO adversary against the Stockfish-PD and Random10M-NT models cannot be explained by the ratio of prematurely ended games, because there are none in these datasets (and, in the former case, PD eliminates premature game ends as well). The other notable exception is the sequence model move (SMM) baseline adversary against the Lichess-NT models, which suggests a degree of overfitting to the errors present in the dataset.

While incorrectly predicting the end of the game is still a rule violation and is enough to show that the implicit world models are not sound, our findings reveal that our adversaries do not solely rely on this error type. Furthermore, even if the adversaries succeed this way, it is not a result of the models overfitting to this type of error in the dataset.

Table 14: Attack success rates achieved by simply letting the models generate the sequence after processing the warmup prefix.

|  | R500k | R2M | R10M | MB500K | SF8M | LC8M |
|---|---|---|---|---|---|---|
| NT | 0.616 | 0.563 | 0.263 | 0.762 | 0.316 | 0.427 |
| PD | 0.947 | 0.966 | 0.921 | 0.935 | 0.936 | 0.916 |
| NT+JP | 0.696 | 0.609 | 0.273 | 0.740 | 0.395 | 0.349 |
| PD+JP | 0.918 | 0.975 | 0.866 | 0.955 | 0.956 | 0.891 |

### E.3 SEQUENCE MODELS FAIL BY THEMSELVES

A further possible baseline adversary against sequence models can be implemented by letting the sequence model simply generate the sequence until it 'fails by itself'. While this method does not conform to our definition of an adversary, it is probably the easiest way to verify the soundness of the implicit world model.

Table 14 presents the adversarial success rates achieved by this simple method. Impressively, this method achieves high ASRs against Lichess-NT models; however, it is generally among the weaker adversaries.

### E.4 DOES THE BSO ADVERSARY SUCCEED?

The goal of the board state oracle (BSO) adversary is to cause the sequence model's associated board state probe to have as many errors as possible when predicting the board state. One could assume that the reason behind the weakness of the BSO adversary is that it fails to cause the probe to have significant errors.

Figure 6 shows the move-wise mean accuracies and piece accuracies under non-adversarial conditions (evaluated on unseen test games), as well as when the BSO adversary is used to generate moves for white. The BSO adversary is able to guide the game towards regions where the probe's accuracy is significantly higher than its error on non-adversarial test games.

Despite its success in inducing errors in the probed board state, BSO still fails to be an effective adversary against the rule-following capabilities of our sequence models. This further shows the limited causal connection between the generative model's function and the board state probe's output.

### E.5 AGREEMENT BETWEEN MODELS AND PROBES

Here, we delve into the agreement between the ground truth board state, the output of the board state probe, and the implicit board state representation of the sequence models. For a move sequence $s \in \Sigma^*$, let us define the implicit world state representation of a sequence model $M$ as $W_M(s) = \{a \in \Sigma : M(a|s) \geq \epsilon\}$, i.e. the set of actions with at least $\epsilon$ conditional probability. Given a world state probe $B$, let us denote the set of legal actions in $B(M, s)$ (i.e., the world state predicted by the probe) as $W_B(s) \subseteq \Sigma$. As introduced in the main text, $W(s) \subseteq \Sigma$ represents the set of legal actions in the true world model after the action sequence $s$.

Let us use the intersection over union (IoU) metric to quantify the agreement between the true world model, the world state probe, and the implicit world state. Formally,

$$\text{IoU}_{W,M}(s) = \frac{|W(s) \cap W_M(s)|}{|W(s) \cup W_M(s)|} \tag{6}$$

denotes the agreement between the true world state and the implicit world state of $M$,

$$\text{IoU}_{W,B}(s) = \frac{|W(s) \cap W_B(s)|}{|W(s) \cup W_B(s)|} \tag{7}$$

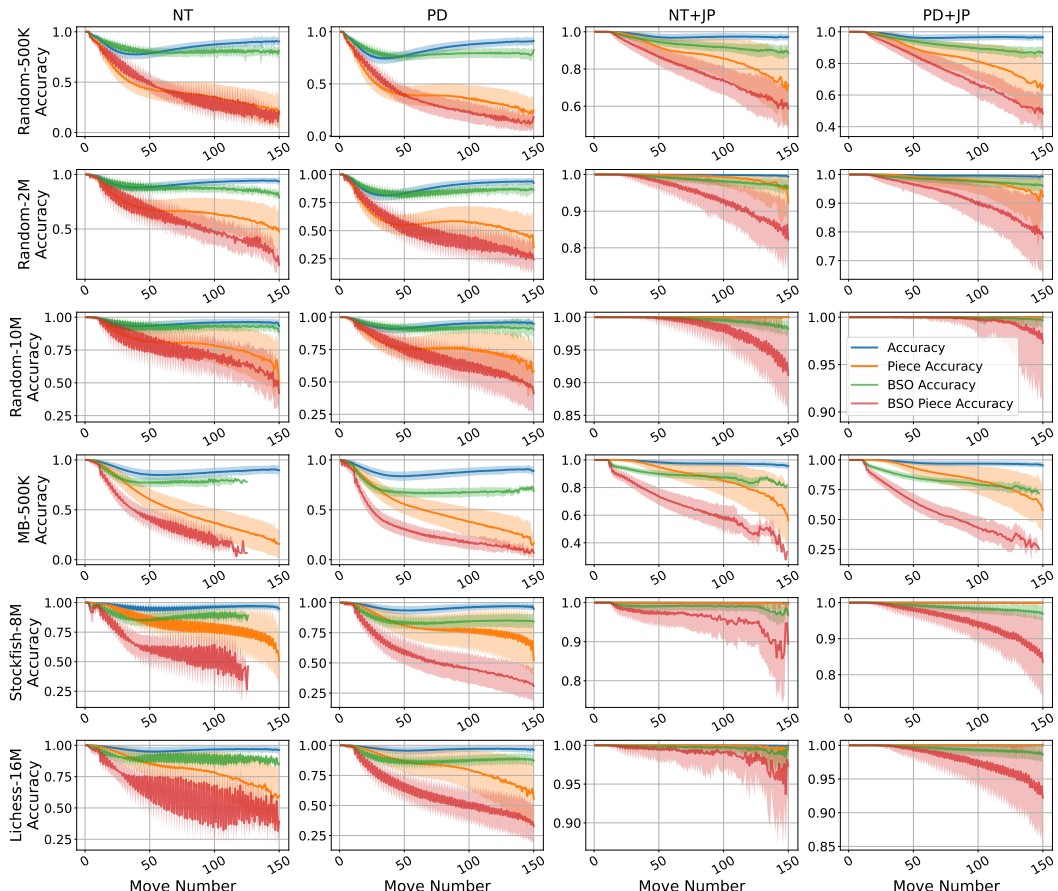

Figure 6: Move-wise mean board state probe accuracies and piece accuracies (similar to Figure 5), along with accuracies and piece accuracies when the BSO adversary generates the moves for white. Note that the BSO adversary takes effect after move 10, which is the end of the warmup sequence. Error bars indicate one standard deviation.

denotes the agreement between the true world state and the state recovered by the world state probe, and

$$\text{IoU}_{M,B}(s) = \frac{|W_M(s) \cap W_B(s)|}{|W_M(s) \cup W_B(s)|} \tag{8}$$

denotes the agreement between the implicit world state of $M$ and the state recovered by the world state probe.

Figure 7 shows the move-wise agreements between the true world model, the board state probes, and the models' implicit world state, evaluated over 15,000 test games that were unseen by either the models or the probes during training. Inspired by Vafa et al. (2024), we used $\epsilon = 0.01$.

Our findings show stark differences between dataset types and training objectives as well. It is clear that models trained on random datasets agree more with the true world state than models trained on curated datasets, as also shown in Li et al. (2023) and Vafa et al. (2024). However, the probability distribution (PD) objective mitigates the probable fragmentation of the NT models throughout all phases of the game, again showing that it is a more effective tool for learning the rules.

More strikingly, there is always a significant difference between the $\text{IoU}_{W,M}$ and $\text{IoU}_{W,B}$, indicating that there is a significant disagreement between the models' next-token predictor heads, and the board states extracted by the probes. This phenomenon is most striking when the next-token prediction and joint probe objectives are combined (NT+JP), where the probes always agree with the true

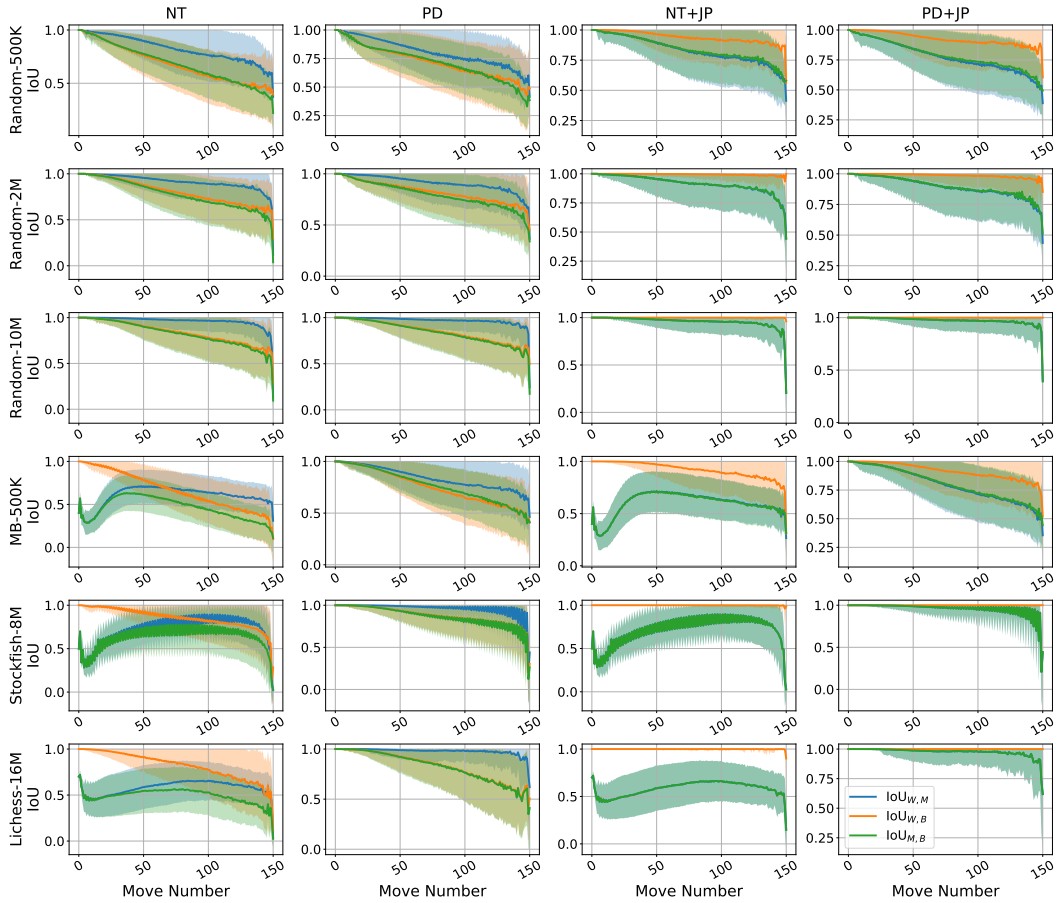

Figure 7: Move-wise mean IoUs between board state probes, model outputs, and the true world state. Error bars represent one standard deviation.

world model, but the agreement between the models and probes, as well as the models and the true world model, is significantly lower.

These results cast further doubt over the causality of probes, as well as the generally accepted probing paradigm, where the probes are trained to extract the ground truth. We believe it would be more beneficial to create probes that directly represent the 'knowledge' of the sequence models, but we leave this up to future work.

### E.6    THE COMPUTATIONAL COST OF OUR ADVERSARIES

Table 15 shows the computational costs of our adversaries against every model. We report these costs in seconds per sequence when using a single H100 GPU, averaged over 1000 sequences used in our evaluation against the top-$k$ sampling strategy. Note that longer sequences yield longer evaluation times.

The cheapest adversary is RMM, as it does not require model inference. The computational costs of SMM and AD are similar as they both require one model inference at each attack step. Interestingly, BSO is computationally inefficient due to the rather costly evaluation of the board state probe, but we admit that our implementation has room for optimization. On the other hand, IMO uses an optimized implementation that predicts the probabilities of single-move continuations using an internal batch size of 128. As expected, IMO is the slowest attack, showing a 10-20× increase in computational cost compared to single-inference attacks like SMM and AD, which is in line with the cost of standard adversarial attacks in other domains.

Table 15: Computational cost of each of our attacks against all models in seconds per sequence, averaged over 1000 sequences.

| | Random-500k | | | | Random-2M | | | | Random-10M | | | |
|---|---|---|---|---|---|---|---|---|---|---|---|---|
| | NT | PD | NT+JP | PD+JP | NT | PD | NT+JP | PD+JP | NT | PD | NT+JP | PD+JP |
| RM | 0.359 | 0.457 | 0.327 | 0.326 | 0.692 | 0.700 | 0.630 | 0.504 | 0.913 | 0.996 | 0.869 | 0.984 |
| SMM | 0.519 | 0.716 | 0.438 | 0.557 | 0.846 | 0.984 | 0.722 | 0.763 | 0.854 | 1.445 | 0.878 | 1.516 |
| IMO | 7.949 | 6.775 | 6.880 | 6.733 | 14.031 | 12.384 | 12.099 | 9.614 | 28.727 | 32.496 | 25.371 | 34.828 |
| BSO | 10.916 | 8.959 | 5.265 | 4.179 | 9.296 | 8.147 | 7.681 | 7.137 | 15.945 | 38.180 | 12.986 | 13.902 |
| AD | 0.655 | 0.590 | 0.604 | 0.514 | 1.281 | 0.866 | 1.097 | 0.777 | 1.513 | 1.650 | 1.586 | 1.640 |
| | Millionbase-500k | | | | Stockfish-8M | | | | Lichess-16M | | | |
| | NT | PD | NT+JP | PD+JP | NT | PD | NT+JP | PD+JP | NT | PD | NT+JP | PD+JP |
| RM | 0.314 | 0.238 | 0.402 | 0.205 | 0.469 | 1.086 | 0.507 | 0.932 | 0.365 | 1.035 | 0.372 | 0.978 |
| SMM | 0.493 | 0.427 | 0.582 | 0.388 | 0.482 | 1.533 | 0.443 | 1.426 | 0.798 | 1.439 | 0.788 | 1.425 |
| IMO | 6.258 | 4.824 | 7.926 | 3.984 | 18.339 | 31.363 | 17.668 | 34.055 | 13.037 | 33.389 | 13.391 | 34.174 |
| BSO | 11.228 | 11.200 | 5.007 | 4.471 | 7.800 | 17.158 | 6.549 | 17.024 | 4.512 | 23.387 | 2.976 | 18.685 |
| AD | 0.567 | 0.341 | 0.673 | 0.287 | 0.796 | 1.524 | 0.766 | 1.535 | 0.609 | 1.644 | 0.579 | 1.549 |

Table 16: Success rate of each attack strategy over all models with the top-$p$ decoding strategy ($p = 0.9$). Results are averaged over three separate evaluations over the same set of warmup sequences. Bold and italic represent the highest and lowest success rates for a model, respectively.

| | Random-500k | | | | Random-2M | | | | Random-10M | | | |
|---|---|---|---|---|---|---|---|---|---|---|---|---|
| | NT | PD | NT+JP | PD+JP | NT | PD | NT+JP | PD+JP | NT | PD | NT+JP | PD+JP |
| RM | 0.983 | 0.993 | 0.990 | 0.996 | 0.936 | 0.969 | 0.955 | 0.980 | 0.804 | 0.911 | 0.833 | 0.908 |
| SMM | *0.951* | 0.997 | *0.966* | 0.995 | *0.805* | 0.968 | *0.861* | 0.984 | *0.408* | 0.921 | *0.451* | 0.925 |
| IMO | **1.000** | **1.000** | **1.000** | **1.000** | **0.998** | **0.998** | **0.998** | **1.000** | **0.978** | **0.965** | 0.975 | **0.962** |
| BSO | 0.974 | *0.982* | 0.975 | *0.980* | 0.928 | *0.964* | 0.933 | *0.964* | 0.784 | *0.878* | 0.816 | *0.873* |
| AD | 0.990 | 0.995 | 0.993 | 0.993 | 0.986 | 0.975 | 0.987 | 0.979 | 0.973 | 0.933 | **0.978** | 0.924 |
| | Millionbase-500k | | | | Stockfish-8M | | | | Lichess-16M | | | |
| | NT | PD | NT+JP | PD+JP | NT | PD | NT+JP | PD+JP | NT | PD | NT+JP | PD+JP |
| RM | 0.972 | 0.998 | 0.967 | 0.998 | 0.383 | 0.931 | 0.379 | *0.918* | 0.254 | 0.917 | 0.183 | 0.913 |
| SMM | 0.967 | 0.999 | 0.970 | 0.999 | *0.226* | 0.946 | *0.248* | 0.943 | 0.507 | 0.933 | 0.459 | 0.915 |
| IMO | **0.999** | **1.000** | **0.998** | **1.000** | **0.832** | **0.981** | **0.836** | **0.980** | **0.667** | **0.974** | **0.649** | **0.976** |
| BSO | *0.953* | *0.992* | *0.944* | *0.990* | 0.404 | *0.909* | 0.441 | 0.919 | 0.226 | *0.873* | 0.178 | *0.893* |
| AD | 0.963 | 0.997 | 0.951 | 0.997 | 0.360 | 0.939 | 0.366 | 0.938 | *0.194* | 0.926 | *0.150* | 0.931 |

## E.7 RESULTS AGAINST TOP-$p$ SAMPLING

Table 16 shows the ASR of our attacks against our models with the top-$p$ sampling policy ($p = 0.9$). These results echo our findings with the top-$k$ sampling policy in Section 8. The success rates of each attack is higher than the ASR against the greedy decoding policy, giving further evidence to the generalizability of our method.

## E.8 TOWARDS ADAPTIVE ADVERSARIES

In this section we present a modification of the Illegal Move Oracle (IMO) adversary that can be seen as an adaptive variant of the IMO variant we used in the main text. As opposed to selecting the move that maximizes the conditional probability of an illegal continuation, this variant aims to find the move that maximizes the sum of the conditional probabilities of all illegal continuations.

In practice, our implementation only analyzes single-move continuations that are reachable by top-$k$ sampling. When we set $k$ to be the size of the vocabulary, the attack is equivalent to the original idea

Table 17: ASR of out adaptive IMO attacks against our models with the greedy decoding strategy (left), and the top-$k$ sampling decoding strategy (right).

| | NT | PD | NT+JP | PD+JP | | NT | PD | NT+JP | PD+JP |
|---|---|---|---|---|---|---|---|---|---|
| R-500K | 1.000 | 1.000 | 0.998 | 1.000 | R-500K | 1.000 | 0.999 | 1.000 | 1.000 |
| R-2M | 0.999 | 0.999 | 0.995 | 0.998 | R-2M | 0.998 | 0.995 | 0.997 | 0.999 |
| R-10M | 0.994 | 0.967 | 0.995 | 0.911 | R-10M | 0.979 | 0.904 | 0.979 | 0.899 |
| MB-500K | 1.000 | 1.000 | 0.998 | 1.000 | MB-500K | 0.999 | 1.000 | 0.994 | 1.000 |
| SF-8M | 0.644 | 0.985 | 0.661 | 0.990 | SF-8M | 0.670 | 0.941 | 0.680 | 0.953 |
| LC-16M | 0.481 | 0.951 | 0.417 | 0.947 | LC-16M | 0.561 | 0.923 | 0.458 | 0.903 |

above. As a result of this modification, this can be seen as an adaptive attack against top-$k$ sampling, although sampling is done on the token level, and the attack analyzes moves (that are made of 2 or 3 tokens).

Table 17 shows the results of this attack against our models with both the greedy decoding strategy and the top-$k$ sampling strategy. Surprisingly, this attack achieved marginally higher ASR against models with greedy decoding (compared to that of of the original IMO in Table 1), and somewhat lower ASR against models with top-$k$ decoding (compared to the success rates in Table 5). This surprising finding hints at a disconnect between the token-level decoding strategies of the models and the move-level analysis of the attacks.

## F  BREAKING DOWN HOW OUR MODELS BREAK DOWN

Here, we investigate the types of errors our models made as a result of our adversarial evaluation. We first provide a taxonomy of possible errors, analyze their frequencies, and provide further fine-grained insights into some of the more complex errors.

### F.1  A TAXONOMY OF ERRORS

Let us start by introducing seven error categories:

(1) **Nonexistent Piece**: The model tries to move a piece that does not exist. In other words, the starting square predicted by the model is empty.

(2) **Opponent's Piece**: The model tries to move a piece that belongs to its opponent. In other words, the starting square predicted by the model contains the opponent's piece.

(3) **Immovable Piece**: The model tries to move a piece that cannot be moved for some reason, e.g., it is blocked, or the model has to block a check and the selected piece is unable to do so, etc.

(4) **Invalid Direction**: The model picks a movable piece, but moves it in an invalid direction, e.g., moving a rook diagonally or a bishop horizontally.

(5) **Erroneous Move**: The error made by the model cannot be categorized into the previous categories, e.g., jumping over pieces, capturing the opponent's king, invalid castling, incorrect promotion, moving the king next to the opponent's king, etc.

(6) **Structural Error**: The move predicted by the model is not in the UCI notation, e.g., the model predicts `e8q` as its move.

(7) **Incorrect End Prediction**: The model incorrectly predicts the end of the game. This error type was analyzed in Appendix E.2.

Note that our taxonomy is by no means a complete breakdown of all possible error types in chess, but it serves as a sensible grouping of the possible failure modes. In addition, not all failure modes can be attributed to an atomic deficiency in the model. Only error types (1) and (2) can be clearly attributed to the model having an incorrect understanding of the board state, but error types (3), (4), (5), and (7) can all arise from an incorrect board state representation, a lack of understanding the rules, or even an incorrect representation of the game history as well.

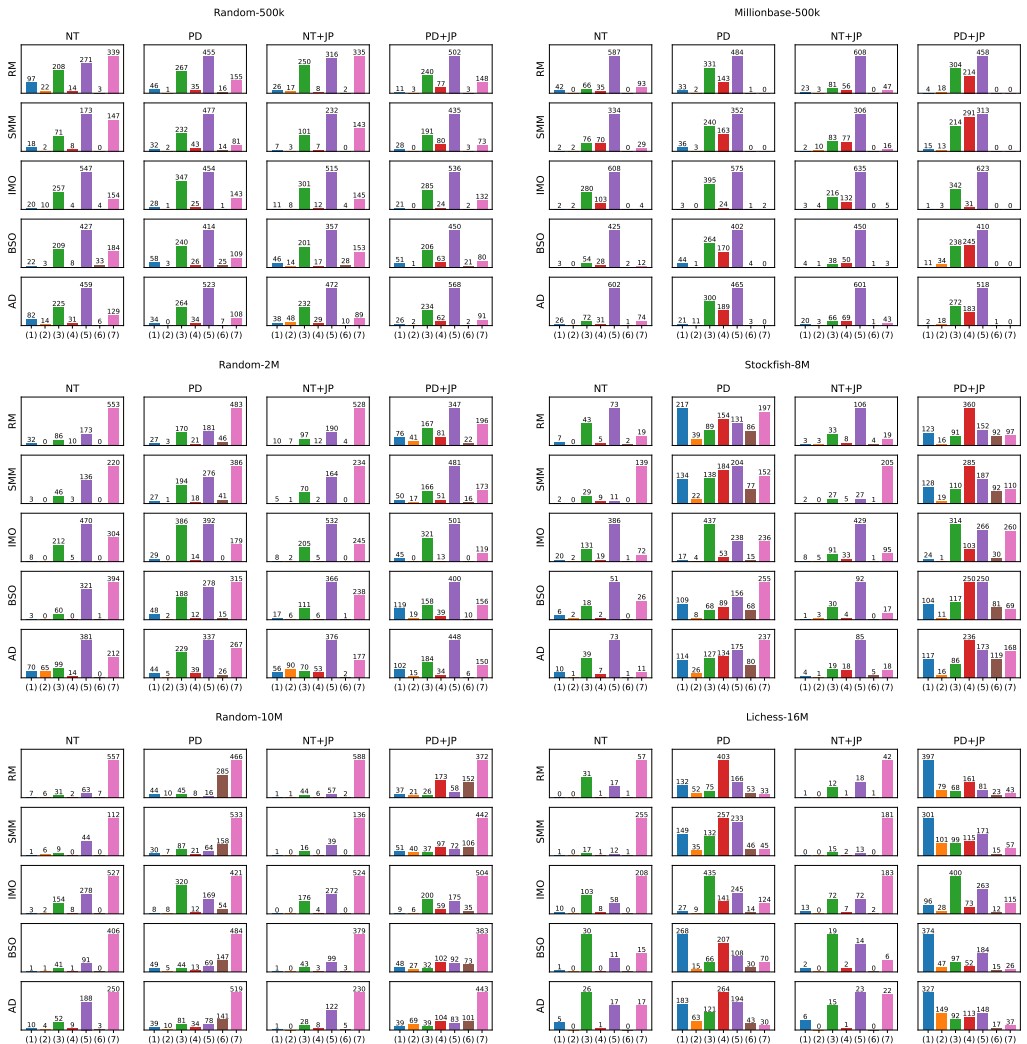

Figure 8: Frequencies of different error types made by our models against all adversaries, with models trained on random datasets being shown on the left, and models trained on curated datasets on the right.

**Results.** Figure 8 shows the frequencies of different error types. Note that our attack does not differentiate between error types; an error type not being prevalent in our evaluation does not mean the model is guaranteed to not make that error, only that other errors are easier to cause.

For all models, Immovable piece (type 3) and erroneous move (type 5) errors are always among the most prevalent. Incorrect end prediction (type 7) is more common for models trained on random datasets, as also demonstrated in Appendix E.2.

The difference between the NT and PD objectives is relatively small when random datasets are used in training, but remarkable when curated datasets are used instead. The PD objective leads to a more uniform error distribution which, when combined with our earlier analysis on model resilience, suggests that PD models fundamentally break down towards the end of the game.

When it comes to attacks, the four weaker attacks (RM, SMM, BSO, and AD) almost always yield similar error distributions. The only exception is SMM against models trained on curated datasets with the NT objective, where it achieves high ASR by causing the model to incorrectly predict the end of the game. However, IMO is clearly different, as it achieves errors related to rule knowledge (types 3, 4, and 5) more frequently.

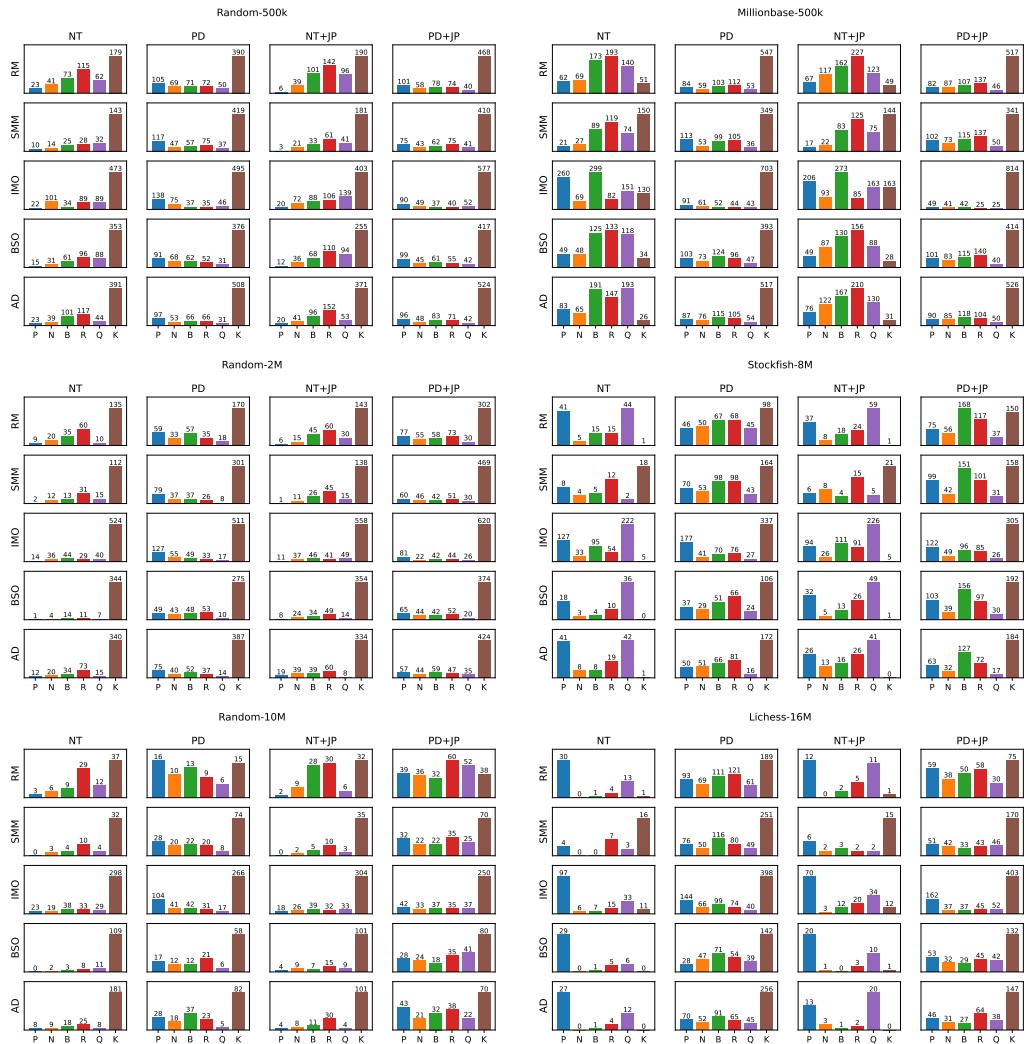

Figure 9: Frequencies of illegally moved pieces grouped by piece type for complex, rule-based errors. Results of models trained on random datasets are on the left, and that of models on trained on are on the right.

## F.2    THE IMPACT OF PIECE TYPES IN COMPLEX ERRORS

We further analyze the impact of piece types in complex errors, namely immovable piece (type 3), invalid direction (type 4), and erroneous move (type 5) from our previous taxonomy. Here, we investigate which pieces the model tries to move, but moves illegally.

Figure 9 shows the results for every model and attack. The trends are largely similar to our earlier analysis on general error types. Models trained on random datasets, as well as those trained with the PD objective overwhelmingly struggle with king moves, while models trained on large curated datasets with the NT objective predominantly struggle with pawn moves.

## G    RESULTS ON LLAMA MODELS

We trained LLaMA models (Touvron et al., 2023) with the settings described in Section 4, resulting in a further 24 models. We used the same architecture size as with the GPT-2 architecture, as described in Section 4.3. We then evaluated them using our adversarial framework, adhering to the settings described in Section 5.

Table 18: Success rate of each attack strategy against LLaMA models. Bold and italic represent the highest and lowest success rates for a model, respectively.

| | Random-500k | | | | Random-2M | | | | Random-10M | | | |
|---|---|---|---|---|---|---|---|---|---|---|---|---|
| | NT | PD | NT+JP | PD+JP | NT | PD | NT+JP | PD+JP | NT | PD | NT+JP | PD+JP |
| RM | 0.944 | 0.987 | 0.961 | 0.990 | 0.899 | 0.935 | 0.885 | 0.910 | 0.703 | 0.839 | 0.680 | 0.811 |
| SMM | *0.777* | *0.892* | *0.792* | *0.950* | *0.702* | 0.940 | *0.645* | 0.930 | *0.285* | 0.863 | *0.286* | 0.877 |
| IMO | **0.999** | **1.000** | **0.999** | **1.000** | **0.999** | **1.000** | **0.999** | **1.000** | **0.979** | **1.000** | **0.980** | **0.990** |
| BSO | 0.885 | 0.953 | 0.896 | 0.951 | 0.826 | *0.875* | 0.814 | *0.873* | 0.525 | *0.699* | 0.581 | *0.772* |
| AD | 0.973 | 0.986 | 0.968 | 0.990 | 0.930 | 0.960 | 0.889 | 0.958 | 0.726 | 0.901 | 0.672 | 0.868 |

| | Millionbase-500k | | | | Stockfish-8M | | | | Lichess-16M | | | |
|---|---|---|---|---|---|---|---|---|---|---|---|---|
| | NT | PD | NT+JP | PD+JP | NT | PD | NT+JP | PD+JP | NT | PD | NT+JP | PD+JP |
| RM | 0.853 | 0.994 | 0.849 | 0.993 | 0.303 | 0.917 | 0.267 | *0.841* | 0.229 | 0.888 | 0.230 | 0.848 |
| SMM | 0.767 | *0.867* | 0.745 | *0.915* | *0.178* | 0.928 | *0.196* | 0.895 | 0.280 | 0.883 | 0.331 | 0.859 |
| IMO | **0.998** | **1.000** | **0.999** | **1.000** | **0.838** | **1.000** | **0.780** | **0.994** | **0.659** | **0.996** | **0.646** | **0.995** |
| BSO | *0.754* | 0.921 | 0.756 | 0.959 | 0.264 | *0.904* | 0.273 | 0.861 | *0.096* | *0.766* | *0.148* | *0.806* |
| AD | 0.846 | 0.999 | 0.843 | 0.992 | 0.298 | 0.925 | 0.232 | 0.892 | 0.194 | 0.863 | 0.179 | 0.852 |

Table 18 shows the success rates of each attack against all 24 LLaMA models, and Figure 10 shows the dynamics of each attack. Notably, all our findings hold true for the LLaMA architecture as well, showcasing that the errors we found with our methodology are not architecture-specific.

Notably, LLaMA models are even less sound than GPT-2 models, with most errors occuring before the 150-move mark. However, as shown in Figure 10, these models also exhibit a substantial bias towards the 150-move sequence length, showcasing that the models pick up irrelevant patterns when it comes to rule learning. Interestingly, LLaMA models can predict legal moves beyond the 150-move mark, which is most notable with models that were trained with the probability distribution (PD) objective, further showcasing that PD facilitates rule learning better than the next-token prediction (NT) objective. We suspect this capability is a result of the LLaMA architecture replacing absolute positional embedding with rotary positional embeddings (Su et al., 2024).

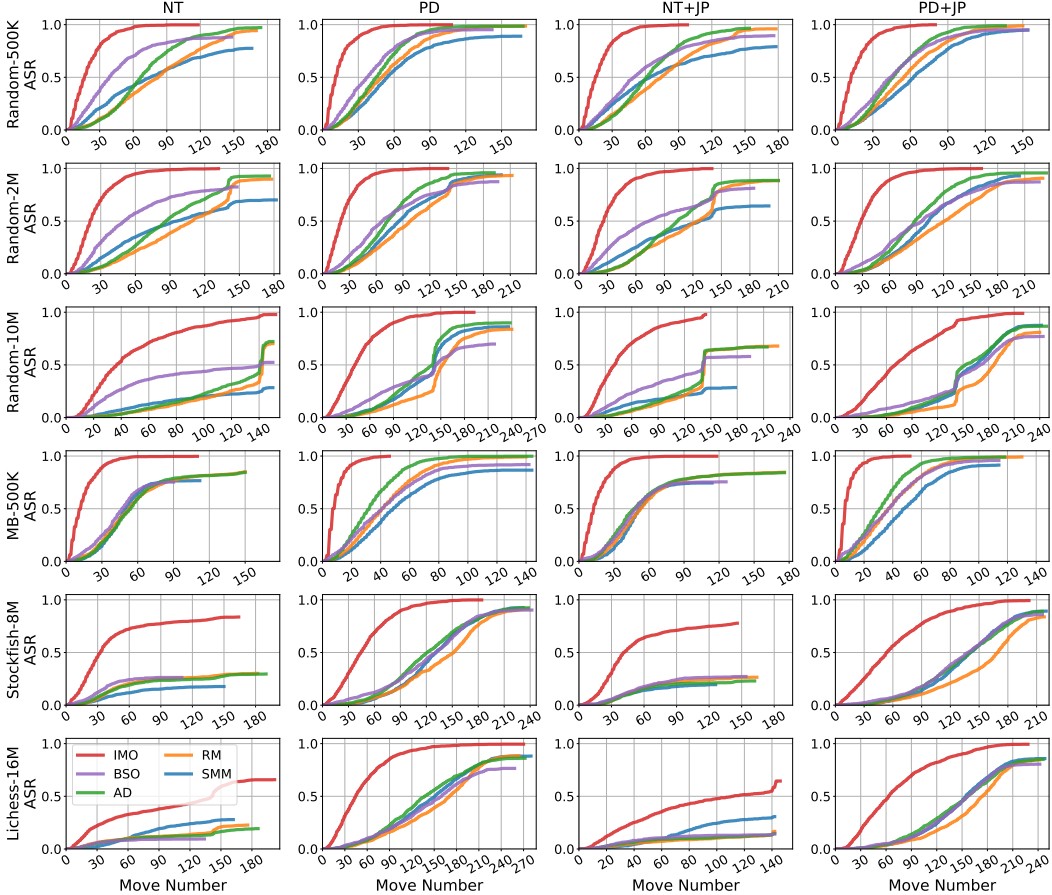

Figure 10: Attack dynamics demonstrated by the move-wise attack success rate (ASR) for each dataset (row) and model (column) using the LLaMA architecture. On each plot, the X-axis shows the move number, and the Y-axis shows the ASR attained by the attacks. Stronger attacks increase ASR more quickly

