# OpenReview forum: "Verification of the Implicit World Model in a Generative Model via Adversarial Sequences"
_ICLR.cc/2026/Conference — ICLR 2026 Poster_

### Official Review · Reviewer_ggS8 · 2025-10-31

**Soundness:** 3
**Presentation:** 4
**Contribution:** 3
**Rating:** 6
**Confidence:** 3

**Summary:**

A new method based on adversary attack is proposed as a way of verifying a generative sequential model's world understanding. Unlike previous methods, this method uses an adversary that plays only legal moves but aims to trigger an illegal next move from the model. The authors test several adversary strategies on gpt2 style chess models trained on different datasets, and find that all models can be broken, implying none are fully sound. Even larger datasets and improved training objectives only delay, not prevent, failure. After all, the result suggests that generative models like GPT still lack understanding of the underlying world rules, despite appearing so.

**Strengths:**

- Paper is well written. Even as someone not fully familiar with the research topic, I could follow it quite well.
- Their proposed verification framework, which focuses on model behaviors, complements previous approaches based on probing, which are more retrospective and analytical.
- The method is effective while remaining easy to implement.
- Their experiments include multiple adversary strategies and a broad range of datasets, giving stronger support to their conclusions.
- The paper also presents insightful diagnostics (eg, probe gradient alignment, OOD warmup tests) that help explain why surface-level rule-following can be misleading.

**Weaknesses:**

- Experiments are based on a rather outdated model (gpt2 with 86M parameters). This leaves open the question of whether stronger world understanding might emerge as models get larger. While the authors mention that their choice was constrained by compute in the last section, I find this justification unconvincing. Given access to h100 GPUs, training newer but still relatively small models (e.g., 2B or 7B) should not pose excessive computational burdens.
- The authors use greedy decoding without much justification. Although greedy decoding is appealing as it maximizes the likelihood of producing a legal move, real chess players (and possibly more human-like models) do not necessarily reason this way. It would be interesting to see whether different sampling strategies lead to different types or frequencies of illegal moves.
- This is not necessarily a weakness, but more of a suggestion: an analysis of common failure modes would have been valuable. For example, does the chance of failure depend on the sequence length, the type of chess pieces involved, or the board state complexity? Identifying such patterns could deepen understanding of how and why the models fail.

**Questions:**

See the above weaknesses.

Besides, it might be interesting to explore whether causal inference/discovery ideas could be applied to this setup (although not necessarily straightforward or even relevant). For instance, one could imagine treating the model’s internal states as potential mediators between the world state and its next prediction, or performing small interventions on hidden features to test causal dependence. If such analyses are possible, they could offer a complementary way to examine whether rule-following behavior arises from genuine causal reasoning or just statistical association.

---

> ### Author Response · Authors · 2025-11-25
> **Response**
>
> We are pleased to see that the reviewer found the paper insightful and easy to follow. Let us answer the questions below.
>
> **Model quality**:  Although the model size we used is not large, the quality of some of our models, when measured using the usual metrics such as next move accuracy, is rather good. Indeed, we have not emphasized this, but now we include detailed information in the Appendix, and mention this in the main paper as well. While models trained on smaller datasets (Random-500k and Millionbase-500k) achieve relatively low legal move ratios between 94.65% and 96.71%, models trained on large datasets (Random-10M, Stockfish-8M, and Lichess-16M) achieve very high legal move ratios between 99.75% and 99.98%. The motivation behind our work is to look beyond these shallow, seemingly favorable metrics, in line with (Vafa et al. 2024). Scaling up the evaluation is always desirable, but running the entire set of experiments at a significantly larger scale is currently beyond our reach. For example, a single 1.4B parameter model on the Lichess-16M dataset takes 10 days to train.
>
> **Sampling strategies**: We ran our set of experiments with the rank-based top-$k$ and probability-based top-$p$ sampling methods and added these results to the paper (see our Common Response for more details). Our methods show a very similar performance under these decoding strategies as well. In fact, our attacks reach higher ASRs against sampling methods than against greedy decoding, including the IMO attack that was specifically designed for greedy decoding. This also means that using greedy decoding to test the soundness of the world model is a reasonable choice.
>
> **Error taxonomy**:  We added an error taxonomy and analysed the common failure modes in the appendix. We found that the error type distribution is markedly different for the different models. We also compared these statistics for not only greedy decoding but other sampling strategies as well, and the failure modes of our models with sampling remain largely the same, although the frequencies of simpler mistakes, such as moving nonexistent or opposing pieces and structural errors, increase slightly. However, complex errors (such as picking an existing piece but making an illegal move) remain dominant in all cases, and the distribution of piece types involved in these errors remains the same.
>
> **Causal inference**: We thank the reviewer for the suggestion on extending this line of work in the future from the perspective of causal inference. We agree that it is an exciting future direction. We believe that our results with board state probes represent a small step in this direction, since our board-state-based attack BSO can be considered as an example of implicit counterfactual reasoning, where we test whether making decisions based on different predicted board-states has the expected causal connection with errors (and we found this connection to be weak).

---

### Official Review · Reviewer_9qiV · 2025-10-31

**Soundness:** 3
**Presentation:** 4
**Contribution:** 3
**Rating:** 6
**Confidence:** 4

**Summary:**

This papers introduces an adversarial framework for verifying the soundness of implicit world models in generative sequence models, building on the work of, for example, Li et al. (2023) and Vafa et al. (2024) and using chess as a testbed. Soundness is defined as whether a model generates only valid sequences according to the true world model's rules --- that is, given any sequence, the model's continuation (generated by greedy decoding) yields a valid sequence under the true world model. The authors key contribution is to construct adversaries that generate valid moves (following a sequence generated by the model) to force the model to predict illegal continuations.
The authors propose several different adversaries to evaluate the soundness of the model. The paper conducts a thorough empirical study training 24 generative sequence models --- exploring the effects of alternative dataset (random games vs. curated datasets) and different training objectives. The authors main findings are:
(1) None of the models are sound—all can be forced to make illegal moves, though the Illegal Move Oracle (IMO) adversary is far more effective than alternatives. IMO picks the legal move that maximizes the conditional probability of an invalid continuation by the opponent.
(2) Dataset size appears to affect soundness --- larger datasets tend to have more sound models across training objectives.
(3) Alternative objectives appears to matter --- the authors argue that the PD objective improves soundness.
(4) The authors argue that board state probes (i.e., the ability to predict board state using the final representation) has limited causal influence on next token predictions.

**Strengths:**

The paper has many nice strengths:

1) The idea of generating adversarial continuations is clever, and a natural way to assess the implicit world model of generative sequence models. The approach provides an ``existence'' proof of unsoundness.
2) The experimental scope of the paper is impressive --- by my read, 24 models trained across 6 datasets of varying sizes and qualities, evaluated with 5 different adversaries. The systematic experiments enable the authors to make interesting statements about how design choices in training might affect soundness (i.e., dataset size, quality and training objectives).
3) The results make interesting progress in reconciling the results in Vafa et al. (2024) (taking a more functional approach to world model evaluation) and well-known works using probes. The results showing that board state probes have limited influence on model predictions is quite nice.
4) The paper is very well-written and easy to follow.

**Weaknesses:**

I use this box to describe weaknesses and make comments that I'd like to see the authors address.

1) The authors criticize Vafa et al. (2024) for using an ad hoc probability threshold to define the generated language by the model, and claiming that their focus on adversaries avoids this. But there are analogous choices that must be made here. The authors focus exclusively on greedy decoding --- why not other choices? Would they perform differently?

2) The definition of world models as valid continuations is presented with little discussion. It would be useful to point out that this is more general than the DFA framework used in Vafa et al. (2024) --- for example, it applies to context-free grammars, pushdown automata. So the adversary framework is potentially more directly applicable than the Myhill-Nerode based metrics in Vafa et al.

3) It was surprising to me how effective RM and SMM were at generating failures. This should merit more investigation, but the authors somewhat gloss over this. For example, SMM achieves 0.943 against Random-2M-PD, meaning the model frequently fails even when generating its own preferred continuations (with only white moves corrected). These results seem to indicate the models are fundamentally poor at chess, not just vulnerable to adversarial attacks. This undermines the interpretation of IMO's success—are we testing world model soundness or just confirming the models don't play chess well?

4) I found the comparison between PD and NT to be insufficiently visible in the main results. Table 1 looks like it shows mixed results to me, and the clearer evidence is buried in the appendix. If this is main result, it would be nice to bring it into the main text.

5) It would be worthwhile to discuss the computational costs of the adversaries. The IMO adversary requires enumerating all illegal continuations and evaluating their probabilities. This is potentially expensive for large action spaces.

6) While the paper establishes that models are unsound, it provides little insight into how and why they fail. What types of illegal moves do models make? Could the illegal moves generated by categorized? E.g., Do models violate basic piece movement rules (e.g., moving pawns backward, bishops diagonally incorrectly) in particular board positions? A taxonomy of errors would reveal whether models learn some aspects of chess better than others. Analogously,  does IMO consistently exploit particular weaknesses (e.g., rare board configurations, endgame positions, complex tactical situations)? Does BSO create specific types of confusion? Understanding which aspects of the world model are fragile would be more informative than aggregate success rates. These sorts of results would allow the authors to connect the paper to recent work emphasizing that generative models learn heuristics/shortcuts.

7) The paper uses only linear probes on the final layer with absolute encoding. Could it be that chess requires different and slightly more complex probes that Othello (in particular, chess has more piece types and special rules so its a more complex game). It would be nice to see that the authors results on probes are not sensitive to the specific linear probes considered.

**Questions:**

Please see my earlier discussion of weaknesses.

---

> ### Author Response · Authors · 2025-11-25
> **Response**
>
> We thank the reviewer for the positive feedback and the thoughtful and constructive comments and suggestions. Let us address the points raised.
>
> **1 (definition of language)**: The approach of Vafa et al. inherently requires the definition of the formal language that is generated by the sequence model as well as the ground-truth world model, while our methodology requires only the knowledge of the ground-truth world model (the rules of chess in this case) but otherwise it is applicable to every sequence model with any sampling strategy without any change. This is because we are focusing on finding counterexamples for the ground-truth model, and we otherwise ignore the generated language (unlike Vafa et al). In this revision, we added an evaluation of probabilistic sampling strategies for generation, and we found that our methods are still effective against our models when the generation uses these probabilistic sampling strategies. For more details on these additional evaluations, please see our Common Response above.
>
> **2 (scope)**: Very good point, thank you, we added this clarification to the paper. Let us note that this more general scope is the direct consequence of the fact that we do not need to define the generated language (as we recall above in the 1st point).
>
> **3 (model quality)**: The quality of some of our models, when measured using the usual metrics such as next move accuracy, is rather good. Indeed, we have not emphasized this, but now we include detailed information in the Appendix, and mention this in the main paper as well. Although models trained on smaller datasets (Random-500k and Millionbase-500k) achieve relatively low legal move ratios between 94.65% and 96.71%, models trained on large datasets (Random-10M, Stockfish-8M, and Lichess-16M) achieve very high legal move ratios between 99.75% and 99.98%.  On some of these good models, RM and SMM do quite poorly, namely on those that were trained on the next token (NT and NT+JP) tasks. Our IMO attack, when combined with out-of-distribution initialization, performs significantly better. This suggests that our methodology is viable even on high-quality models.
>
>
> **4 (NT vs PD)**: We are pleased that the reviewer finds the results in the Appendix interesting. Given the limited space in the main part of the paper, and after carefully weighing which new results to include there, we ultimately decided that we’d rather keep these results about the difference between NT and PD in the Appendix. This is because we find it more important to emphasize that even PD models are unsound, regardless of their resilience against weaker attacks. This shows that having access to even the ground truth distribution isn't enough to produce sound world models, which is the finding we aimed to highlight.
>
> **5 (cost of adversaries)**: We measured the computational cost of all adversaries and added this to the Appendix as Section E.6. While this depends on many factors (e.g., the number of moves per sequence, the number of legal moves in each position and the internal batch size in IMO), we found IMO to be 10-20 times more expensive than standard generation-based strategies like SMM and AD, which is in line with the computational costs of adversarial evaluation in other domains (e.g. image classification).
>
> **6 (failure patterns)**: In the Appendix, in Section F, we added a comprehensive error-type analysis for all the models and attacks. One question is whether the errors indicate a lack of rule knowledge or an incorrect board state representation, or both? We found that, while models do make simple mistakes (e.g., trying to move nonexistent pieces, which implies an incorrect board state), they mostly make complex errors that cannot be explained by the lack of rule knowledge or an incorrect board state representation alone. The four weaker attacks (RM, SMM, BSO, and AD) all cause the models to make similar mistakes, while IMO more frequently causes errors that could be attributed in part to incorrect rule knowledge. We further analyzed these complex errors and found that most of these are made with the king, but models trained on curated datasets with the NT objective struggle with pawn moves instead, which suggests that at least these moves are based on some set of heuristics.
>
> **7 (linear probes)**: With probes, the general consensus is that more complex, non-linear probes could learn the probing task by themselves by incorporating information that is not present in the input representation. Nevertheless, in our preliminary experiments, we tried both nonlinear probes and linear probes with more complex architectures (in our case, Conv1D probes over the last $k$ tokens), as well as probes with side-specific encodings, and found that they performed similarly to linear probes in our +JP training setting, as well as our adversarial evaluation.

---

### Official Review · Reviewer_oiiD · 2025-11-01

**Soundness:** 3
**Presentation:** 4
**Contribution:** 3
**Rating:** 8
**Confidence:** 3

**Summary:**

This paper belongs to a line of work evaluating the fidelity of a language model in uncovering the hard rules of its underlying environment. The specific strategy in this paper is to adversarially design sequences where the model will incorrectly predict the next token (for instance, the max probability next token is not a valid move in the underlying game). This differentiates itself from other methods which directly measure next-token prediction, are based on linear probes or other white-box methods, or recent work which measures similarity between the model and the DFA-structure of the underlying world model.

This paper focuses on the game of chess, where the adversary is literally playing with the model and attempting to force the model to make an illegal move.
The attacker is greedy and chooses its acception at each timestep according to one of the following rules:
 - IMO: Maximize the conditional probability of the model on an incorrect token.
 - BSO: Maximize the error the board state as given by a linear probe.
 - AD: Choose the token with minimum probability under the model.
 - RM: Random token.
 - SMM: Choose the token with maximum probability under the model. This is standard argmax decoding.

 Three training strategies are employed:
  - NT: next-token prediction, self-supervised over the given dataset
  - PD: also self-supervised, but the target distribution is uniform over all valid moves
  - JP: the model is jointly trained to produce a good board state via a linear probe

Across datasets, training strategies, and attack strategies, the attack success rate can vary significantly. For instance, on a dataset of 500k random moves, RM, IMO, and AD all achieve >90% attack success across all training strategies. On the other hand, on a dataset of 10M real games, the best attack (IMO) only achieves <40% success. Regardless, the attacks, especially IMO, are commonly capable of finding invalid sequences. Interestingly, next-token prediction (NT) is often the best strategy, with PD and JP not adding much benefit or making attacks easier.

The IMO attack is the most robust: all other attacks reduce to small success rates in some settings. Both in terms of attack and training procedure, introducing linear probes does not lead to better performance.

Interestingly, the attacks performed worse on curated datasets than purely random datasets. Prior works showed random datasets have better world model recovery according to their metrics. The authors demonstrate that injecting an out-of-distribution prefix before running the attack significantly increases the attack success rate, demonstrating a limitation of the proposed attacks.

**Strengths:**

- This paper provides a clean, new method of evaluating the ability of the a model to recover the underlying rules of its environment. The attack-based evaluation can be performed on any black-box model and is easy to understand. It avoids having to deal with many subtleties of model inference or choosing a background distribution that appear in other works.
- The experimental setup evaluates several natural and interesting choices of attack, training objective, and training data.
- The paper is clearly written. The takeaways are convincing and laid out well.

**Weaknesses:**

- It is not clear how the success rate is calculated, see the first question.

- While easy to implement and generally applicable, notions like success rate of attacks are not very interpretable: they are a loose proxy. Does a model which has attack success rate 40% rather than 50% understand the underlying world model better? It is probably not that informative at that scale as the success rate depends heavily on the specific attack that is chosen.

- The setup and experiments are all ran with deterministic argmax decoding. While this is a reasonable choice for finding invalid sequences, it would be interesting to investigate more realistic settings where tokens are sampled from the model.

- It would be interesting to consider attacks which are not completely greedy. For instance, they could unravel $k$ steps of the game tree for small $k$. The takeaway from Section 7 suggests that an attack like AD should succeed but perhaps it does not work well in the purely greedy setting.

**Questions:**

- What is the success rate averaged over? As both the model decoding and the adversary are deterministic, how is a percentage success rate achieved?

 - Why isn't IMO defined as the sum over all illegal tokens of their conditional probabilities? Perhaps there is a board state where a single illegal move is unlikely but in aggregate, the model places high probability on illegal moves. I would guess this is related to argmax decoding.

 - On random data, isn't it the case that NT and PD are essentially equivalent (at least in expectation)? This is not fully realized in the numbers, and I do not understand why.

---

> ### Author Response · Authors · 2025-11-25
> **Response**
>
> We are delighted that the reviewer found our paper clear and convincing. Let us address the comments and suggestions below.
>
> **Success rate calculation**: We use 1000 unique prefixes to get different sequences even with deterministic models and adversaries, as detailed in the first paragraph of Section 5. The reported ratios are calculated over these different sequences.
>
> **Sampling strategies**: We ran our set of experiments with the rank-based top-$k$ and probability-based top-$p$ sampling methods and added these results to the paper (see our Common Response for more details). Our methods show a very similar performance under these decoding strategies as well. In fact, our attacks reach higher ASRs against sampling methods than against greedy decoding, including the IMO attack that was specifically designed for greedy decoding. This also means that using greedy decoding to test the soundness of the world model is a reasonable choice.
>
> **Q2 (IMO implementation)**: Yes, it is related to argmax decoding. However, we implemented a variant similar to the one proposed by the reviewer. This made sense, especially because we also evaluated our attacks over probabilistic sampling-based sequence generation (as mentioned above), and the proposed implementation seemed more fitting at first. This new variant takes the probabilities of the top-$k$ moves after each possible move and picks the move that maximizes the probability sum of illegal moves among the top-$k$ moves (which, for $k$=”vocab size”, is equivalent to the reviewer’s idea). We evaluated this attack against models with argmax and top-$k$ decoding, where both the attack and the defense used $k$=4, and added these results to the Appendix. Surprisingly, this attack achieved marginally higher ASR against argmax decoding, and somewhat lower ASR against top-$k$ decoding  (we expected the opposite in both cases, although the differences are not large). This surprising result suggests a disconnect between the token-level decoding strategy of the model and the move-level analysis of the attack. In any case, it is an interesting future direction to understand how IMO strategies and decoding strategies interact.
>
> **Q3 (NT vs PD)**: Over the random datasets, and in expectation, yes, NT and PD are equivalent. In practice, they are not, because in the case of NT, a game-state the model sees will typically have only one continuation in the dataset (except during the very first moves). The branching factor in chess is large, so in random sequences, game states do not tend to repeat. For example, there are over 119 million distinct sequences for only the first six moves of chess. This means that the finite sample used by the NT task to approximate the true distribution (PD) is too small; it is not a good-enough approximation, despite the relatively large dataset sizes.
>
> **Interpretability of ASR**: The quantitative results are not necessarily informative; however, the differences between different attack types are, which helps us choose the best attack. This is important because a theoretically perfect attack would be quantitatively interesting as well. This can only be approximated by using the strongest attack one has access to, since the perfect attack would be prohibitively expensive in practice.
>
> **Non-greedy attacks**:  We agree that attacks with longer horizons and/or more attacked steps would be interesting to study, perhaps along the lines proposed by the reviewer. It is a very interesting and challenging open question whether one can propose significantly better attacks that are not prohibitively expensive.

---

### Official Review · Reviewer_c2oN · 2025-11-01

**Soundness:** 3
**Presentation:** 3
**Contribution:** 3
**Rating:** 6
**Confidence:** 4

**Summary:**

The paper investigates whether sequence models trained on chess possess a sound implicit world model by searching for legal prefixes that cause the model to predict an illegal move.
Several adversaries are introduced: Random Move (RM), Sequential Mimic (SMM), Approximate Decoder (AD), Board State Oracle (BSO), and Illegal Move Oracle (IMO).
Across 24 GPT-2-style models trained with varied datasets and objectives, all models fail the soundness test, although larger datasets and probability-distribution training improve robustness.

**Strengths:**

1. The paper tackles an interesting and clearly defined problem, formulating world-model verification as falsification through adversarial legal sequences.
2. The chosen adversaries (RM, SMM, AD, BSO, IMO) form a meaningful distribution of strength and show how more powerful attacks reveal hidden inconsistencies in the model’s behaviour.
3. The paper is generally clearly written and has a fairly detailed analysis of performance patterns.

**Weaknesses:**

1. It is difficult to interpret the results without understanding how competent these GPT-2-based models are as chess players. A comparison to specialised chess language models or an Elo-style strength estimate would help determine whether the observed unsoundness is surprising or simply reflects limited gameplay skill.
2. The strong dependence on sequence length suggests that the models struggle with long-range reasoning. This is an interesting finding, but it mostly reflects the limitations of the training setup rather than providing deep insight into world-model soundness.
3. Games longer than 150 moves are removed rather than truncated. Truncation could have preserved more board diversity while controlling length. A short justification for this choice would improve clarity.
4. The Illegal Move Oracle (IMO) is predictably the strongest adversary because it has access to a legality oracle. The statement in line 342 that this “showcases the need for strong adversaries to reliably verify generative models” feels too general. In more complex domains where such oracles cannot be built, this approach may not scale.
5. In figure 1, not all plots are complete.
6. Line 139 states that only some algorithms use a board-state decoder, whereas line 304 says every model has a board-state probe. This inconsistency is minor but worth clarifying so readers understand that the probe is used for post-hoc analysis rather than during training.

**Questions:**

1. How competent are the models at playing chess overall? For example, have you measured legality rates or gameplay strength relative to an engine or other chess language models?
2. Do you terminate evaluation at the first illegal move or continue generation afterwards?

---

> ### Author Response · Authors · 2025-11-25
> **Response**
>
> We thank the reviewer for the encouraging comments about the relevance and clarity of our work. Let us address each question and weakness:
>
> **Q1 (model quality)**: We measured the legality rates of all our models and added these results to the Appendix, and we now also mention these briefly in the main text. The lowest legal move ratio achieved by our models was 94.65%, but models trained on larger datasets (Rand-10M, Stockfish-8M, Lichess-16M) achieve a legality rate between 99.75% and 99.98%. This highlights the fact that gameplay quality and legality rate are not strongly tied.
>
> **Q2 (termination of evaluation)**: We terminate the evaluation at the first illegal move. Technically speaking, illegal sequences do not have legal continuations at all, so after an illegal move, the attacks have fulfilled their purpose.
>
> **1 (model gameplay quality)**: This comment touches on an intriguing aspect. Note, however, that our work focuses on the rule learning aspect of soundness, which is a much easier problem than achieving or testing high-quality gameplay. Models that are trained on high-quality gameplay data are inherently stronger players than models that play randomly, but they are not necessarily better at producing legal moves. Our adversaries do not aim to beat these models in chess; instead, they try to force them to make moves that are against the rules of chess. We agree that the interplay between gameplay quality and world model soundness would be interesting to study further. Our discussion of out-of-distribution prefixes in Section 7 is a step in this direction.
>
> **2 (dependence on sequence length)**: We agree with the observation. However, all sequence model training setups suffer from the limitation that training sequences are limited in length, so similar weaknesses are expected for all sequence models, even if they are architecturally designed for better long-range reasoning. To empirically investigate this hypothesis, we present new results in the Appendix with LLaMA models that are considered better long-range reasoners. Our new results demonstrate that, while LLaMA models are sometimes better beyond 150 moves, they still exhibit a strong length bias.
>
> **3 (removal vs truncation)**: While the truncation of long games is also an option, we decided to remove longer games from curated datasets (Stockfish-8M and Lichess-16M) to comply with the framework of Toshniwal et al. As these games represented less than 0.2% of the initial datasets, their added contribution to the quality of our models would not have been significant.
>
> **4 (requirements of an oracle)**: The availability of an oracle is an intrinsic assumption in a verification setting. Our motivation was to offer a methodology for measuring how well sequence models are able to learn rules from data, which we cannot verify without the knowledge of said rules. Also, any knowledge we assume to have can also be used as part of our adversaries, as this does not introduce extra assumptions.
>
> In this work, we assumed full knowledge; however, this assumption could be relaxed, and we could rely on partial knowledge as well. The attacker could then try to generate a conflict with this partial knowledge, but we might miss some illegal sequences generated by the model that are not in conflict with the available partial knowledge.
>
> **5 (presentation)**: The lines end when each attack achieves its highest success rate (the plots are easier to read this way, because the remaining missing parts are just horizontal lines that might cover more interesting other lines, and make the plot look convoluted). We added this clarification to the main text.
>
> **6 (presentation)**: Thank you, we clarified this in the main text.

---

### Author Response · Authors · 2025-11-25
**Common Response**

We thank the reviewers for their thorough reviews, constructive comments, and suggestions. We are pleased to have received positive comments on the relevance, contributions, and clarity of our paper, and the depth of our experimental evaluation.

We revised the submission based on the numerous comments and suggestions of the reviewers. Along with minor clarifications in the main text, the following experimental results were added to the paper.

**Attacks against sampling-based decoding strategies** (suggested by reviewers oiiD, 9qiV, ggS8): We performed our adversarial evaluation using our models with two sampling techniques (top-$k$ and top-$p$). Our conclusions remain unchanged; in fact, our attacks achieve higher ASR against these sampling strategies. We added the results against the top-$k$ strategy to the main text (Section 8), and those against the top-$p$ strategy to the Appendix (Section E.7).

**Error taxonomy and analysis** (suggested by reviewers 9qiV, ggS8): We added a detailed analysis of the errors made by our models during our adversarial evaluations to the Appendix (Section F).

**Legality ratio** (suggested by reviewers c2oN and 9qiV): We analyzed the legal move ratio of our models to establish that, according to this surface-level metric, they are competent at playing legal moves at a rate up to 99.98%. While this does not reflect world model soundness, we now mention this in the main text and added the full breakdown in the Appendix (Section D).

**More modern setup** (suggested by reviewers c2oN and ggS8): Using the same setup, we trained similarly sized LLaMA models and evaluated them with our adversaries. While these models are better at long-range reasoning, as indicated by their performance against our weaker attacks, they still exhibit the sequence-length bias we uncovered in the main text, and are still shown to be unsound by our framework. We added these results to the Appendix (Section G).

**Computational overhead** (suggested by reviewer 9qiV): We analyzed the computational overhead of our attacks and added it to the Appendix (Section E.6).

**Alternative IMO implementation** (suggested by reviewer oiiD): We implemented the alternative IMO attack suggested by the reviewer as an adaptive attack and added its results to the Appendix (Section E.8).

---

### Comment · Area_Chair_mLQb · 2025-11-27
**Reviewer Reminder: Author Rebuttals Available**

Dear Reviewers,

The authors have posted their rebuttals to your reviews.

Please read the authors' responses, assess whether your concerns have been addressed, and update your ratings accordingly.

Your prompt attention to the rebuttals is appreciated.

Best,
AC

---

### Meta-Review · Area_Chair_F8EB · 2026-01-05

**Summary:**

This paper studies the verification of the implicit world model in a generative model in the context of Chess. All reviewers find the studied setting novel and the results provide new insights. The authors’ rebuttal has successfully addressed the major concerns of reviewers. Overall, I recommend acceptance of this submission.

In particular, based on my understanding, the following primary concerns from the initial version have been properly addressed:

- additional attack results and ablation studies as suggested by reviewers
- further analysis on failure modes
- results on newer models

**Reviewer Concerns:**

I believe all concerns were properly addressed

**Reviewer Scores:**

Likely to maintain or increase

---

### Decision · Program_Chairs · 2026-01-26

Accept (Poster)